

# Spectral Intensity Bioaerosol Sensor (SIBS):
# A new Instrument for Spectrally Resolved Fluorescence Detection
# of Single Particles in Real-Time
Tobias Könemann[1], Nicole Savage [2a], Thomas Klimach[1], David Walter[1], Janine Fröhlich-
Nowoisky[1], Hang Su[1], Ulrich Pöschl[1], J. Alex Huffman[2], and Christopher Pöhlker[1]
*[1]Max Planck Institute for Chemistry, Multiphase Chemistry Department, P.O. Box 3060, D-55020*
*Mainz, Germany*
*[2]University of Denver, Department of Chemistry and Biochemistry, 2190 E. Iliff Ave., Denver, Col-*
*orado 80208, USA*
*[a] Now at Aerosol Devices Inc., 430 North College Avenue # 430, Fort Collins, Colorado 80524,*
*USA*
*Correspondence to:* J. A. Huffman (alex.huffman@du.edu) and C. Pöhlker (c.pohlker@mpic.de)
Keywords: SIBS, WIBS, Bioaerosols, Single Particle Fluorescence, Fluorescence Spectroscopy, Per-
formance Evaluation, Polystyrene Latex Spheres, Biofluorophores





**Abstract**
Primary biological aerosol particles (PBAP) in the atmosphere are highly relevant for the Earth sys-
tem, climate, and public health. The analysis of PBAP, however, remains challenging due to their
high diversity and large spatiotemporal variability. For real-time PBAP analysis, light-induced fluo-
rescence (LIF) instruments have been developed and widely used in laboratory and ambient studies.
The interpretation of fluorescence data from these instruments, however, is often limited by a lack of
spectroscopic information. This study introduces a new instrument – the Spectral Intensity Bioaero-
sol Sensor (SIBS) – that resolves fluorescence spectra for single particles and, thus, promises to ex-
pand the scope of fluorescent PBAP quantification and classification.
The SIBS shares key design components with the latest versions of the Wideband Integrated Bioaer-
osol Sensor (WIBS) and the findings presented here are also relevant for the widely deployed WIBS-
4A and WIBS-NEO as well as other LIF instruments. The key features of the SIBS and findings of
this study can be summarized as follows:
- Particle sizing yields reproducible linear responses for particles in the range of 300 nm to 20 µm.
The lower sizing limit is significantly smaller than for earlier commercial LIF instruments (e.g.,
WIBS-4A and the Ultraviolet Aerodynamic Particle Sizer (UV-APS)), expanding the analytical
scope into the accumulation mode size range.
- Fluorescence spectra are recorded for two excitation wavelengths ($\lambda_{ex}$ = 285 and 370 nm) and a
wide range of emission wavelengths ($\lambda_{mean}$ = 302 – 721 nm) with a resolution of 16 detection
channels, which is higher than for most other commercially available LIF bioaerosol sensors.
- Fluorescence spectra obtained for 16 reference compounds confirm that the SIBS provides suffi-
cient spectral resolution to distinguish major modes of molecular fluorescence. For example, the
SIBS resolves the spectral difference between bacteriochlorophyll and chlorophyll *a/b*.
- A spectral correction of the instrument-specific detector response is essential to use the full fluo-
rescence emission range.
- Asymmetry factor (AF) data were assessed and were found to provide only limited analytical
information.
- In test measurements with ambient air, the SIBS worked reliably and yielded characteristically
different spectra for single particles in the coarse mode with an overall fluorescent particle frac-
tion of ~4 % (3σ threshold), which is consistent with earlier studies in comparable environments.



## 1.    Introduction

Aerosol particles are omnipresent in the atmosphere, where they are involved in many environmental and biogeochemical processes (e.g., Baron & Willeke, 2001; Després et al., 2012; Fuzzi et al., 2006; Hinds, 1999; Pöschl, 2005; Pöschl & Shiraiwa, 2015). Primary biological aerosol particles (PBAP), also termed bioaerosols, represent a diverse group of airborne particles, consisting of whole or fragmented organisms including, e.g., bacteria, viruses, archaea, algae, and reproductive units (pollen and fungal spores), as well as decaying biomass (e.g., Deepak & Vali, 1991; Després et al., 2012; Fröhlich-Nowoisky et al., 2016; Jaenicke, 2005; Madelin, 1994; Pöschl, 2005) and can span sizes from few nanometers up to ˜100 µm (Hinds, 1999; Schmauss and Wigand, 1929). The increasing awareness of the importance of PBAP regarding aerosol-cloud interactions, health aspects, and spread of organisms on local, continental or even intercontinental scales has led to a growing interest by scientific researchers and the public (e.g., Després et al., 2012; Fröhlich-Nowoisky et al., 2016).

Due to inherent limitations (e.g., poor time resolution and costly laboratory analyses) of traditional off-line techniques (e.g., light microscopy and cultivation-based methods) for PBAP quantification, several types of real-time techniques have been developed within the last several decades to provide higher time resolution and lower user costs (e.g., Caruana, 2011; Després et al., 2012; Fennelly et al., 2017; Ho, 2002; Jonsson & Tjärnhage, 2014; Sodeau & O'Connor, 2016). One promising category of real-time instruments – meaning that particles are sampled and analyzed both instantly and autonomously – involves application of light- induced fluorescence (LIF). The main principle of this technique is the detection of intrinsic fluorescence from fluorophores ubiquitous in biological cells, such as those airborne within PBAP. These fluorophores include a long list of biological molecules such as aromatic amino acids (e.g., tryptophan and tyrosine), co-enzymes (e.g., reduced pyridine nucleotides (NAD(P)H)), flavin compounds (e.g., riboflavin), as well as biopolymers (e.g., cellulose and chitin) and chlorophyll (e.g., Hill et al., 2009; Li et al., 1991; Pan et al., 2010; Pöhlker et al., 2012, 2013). Detailed information of biological fluorophores can be found elsewhere (Pöhlker et al., 2012 and references therein).

Today, commercial on-line LIF instruments such as the Ultraviolet Aerodynamic Particle Sizer (UV-APS, TSI Inc. Shoreview, MN, USA) and the Wideband Integrated Bioaerosol Sensor (WIBS, developed at the University of Hertfordshire, U.K. and currently licensed and manufactured by Droplet Measurement Technologies (DMT, Longmont, CO, USA)) are commonly applied for research purposes. Detailed descriptions of the UV-APS (e.g., Agranovski et al., 2003; Brosseau et al., 2000; Hairston et al., 1997) and the WIBS series (e.g., Foot et al., 2008; Kaye et al., 2000, 2005; Stanley et al., 2011) are given elsewhere. Concisely, the UV-APS uses an $\lambda_{ex}= 355$ nm laser excitation source and spans an emission range between $\lambda_{em}= 420\text{-}575$ nm. In contrast, the WIBS applies two pulsed xenon flash lamps emitting at $\lambda_{ex}= 280$ and $370$ nm, whereas fluorescence emission is detected in



three detection channels, $\lambda_{em}$= 310 - 400 nm (at $\lambda_{ex}$= 280 nm) and $\lambda_{em}$= 420 - 650 nm (at $\lambda_{ex}$= 280 and
370 nm). Both instruments provide spectrally unresolved fluorescence information. The latest WIBS
model is currently the WIBS-NEO, whose design is based on a WIBS-4A but with an extended par-
ticle size detection range between ~300 nm and 100 µm (nominal). Both UV-APS and WIBS models
have been examined in a variety of laboratory validations (e.g., Agranovski et al., 2003, 2004;
Brosseau et al., 2000; Healy et al., 2012; Hernandez et al., 2016; Kanaani et al., 2007; O'Connor et
al., 2013; Saari et al., 2013, 2014; Savage et al., 2017; Toprak & Schnaiter, 2013) and have been
deployed to investigate both indoor and outdoor atmospheric aerosol via longer-term measurements
(e.g., Bhangar et al., 2014; Crawford et al., 2015; Fernández-Rodríguez et al., 2018; Foot et al., 2008;
Gabey et al., 2010, 2013; Gosselin et al., 2016; Healy et al., 2014; Huffman et al., 2010, 2012, 2013;
Perring et al., 2015; Schumacher et al., 2013; Twohy et al., 2016; Ziemba et al., 2016).

Although LIF instruments do not offer the same ability to qualitatively identify sampled particles

as, e.g., off-line microscopy, mass spectrometry, or culture-based methods, they provide size-re-
solved information as well as fast sampling and fine-scale temporal information for single particles
not accessible with off-line techniques. Nevertheless, these instruments present significant chal-
lenges. For example, quantification of PBAP by LIF instruments is hindered by the fact that some
biological materials reveal weak fluorescence characteristics that does not rise above detection
thresholds (Huffman et al., 2012). In addition to this complication, the detection threshold is not a
universally defined parameter and varies for each channel between different units of the same type
of instruments (e.g., Hernandez et al., 2016; Savage et al., 2017). Furthermore, unambiguous spec-
troscopic characterization of bioparticles is fundamentally challenging, because fluorescence spectra
of even individual molecules in condensed matter are relatively broad due to radiative decay path-
ways of excited electrons. Further, bioparticles are chemically complex, each comprised of a mixture
of dozens of types of fluorophores that can each emit a unique emission spectrum that smears together
into an even broader fluorescence spectrum from each particle (Hill et al., 2009, 2015; Pan, 2015).
Another difficulty is that many non-biological particles, such as certain mineral dusts and polycyclic
aromatic hydrocarbons (PAHs), may fluoresce, making it more difficult to distinguish patterns aris-
ing from biological particles (e.g., Pöhlker et al., 2012 and references therein; Savage et al., 2017).
Lastly, most currently available commercial LIF instrumentation are limited to recording data in 1-3
spectrally integrated emission channels, which limits the interpretation of fluorescence information.
Recent efforts to apply more complex clustering algorithms to the spectrally unresolved WIBS-type
data are proving helpful at adding additional discrimination, but aerosol characterization using in-
strumentation with such low spectra resolution is likely to be fundamentally limited (e.g., Robinson
et al., 2013; Ruske et al., 2017; Savage & Huffman, 2018).



The evolution of LIF techniques over the last several decades has significantly expanded our
knowledge on spatiotemporal patterns of PBAP abundance in the atmosphere. Nevertheless to further
improve the applicability of LIF instrumentation to widespread PBAP detection, it is necessary both
to design LIF instrumentation with adequate instrumental properties (e.g., high spectral resolution)
and to standardize their operation by characterizing instruments thoroughly with known standards
(Robinson et al., 2017). Working toward this goal, a number of LIF instruments that offer analysis
of single bioparticles by providing resolved fluorescence spectra have been developed (e.g., Hill et
al., 1999; Pan et al., 2003, 2010; Pinnick et al., 2004; Ruske et al., 2017), however relatively little
has been done to offer these commercially. Examples for commercially available instruments provid-
ing resolved fluorescence spectra are the PA-300 ($\lambda_{ex}$= 337 nm; $\lambda_{em}$= 390 – 600 nm, 32 fluorescence
detection channels) (Crouzy et al., 2016; Kiselev et al., 2011, 2013) and the follow-up model Rapid-
E ($\lambda_{ex}$= 337 nm; $\lambda_{em}$= 350 – 800 nm, 32 fluorescence detection channels) (http://www.plair.ch/), both
manufactured by Plair SA, Geneva, Switzerland. Beside of resolved fluorescence detection, both
instruments also provide measurements of the decay of fluorescence signals, also referred to as flu-
orescence lifetime.
Introduced here is a new instrument for the detection and characterization of individual particles;
the Spectral Intensity Bioaerosol Sensor (SIBS, Droplet Measurement Technologies). Technical
properties of the instrument are described in detail and its performance is validated with sizing and
fluorescence particle standards, as well as with ambient air. Due to the dual excitation and spectrally
resolved fluorescence in combination with a broad size detection range, the SIBS has the potential to
increase the selectivity of fluorescent biological and non-biological particle detection and discrimi-
nation. Because the SIBS uses a comparable optical system as the WIBS-4A and WIBS-NEO, tech-
nical details presented here are broadly important to a growing community of scientists investigating
both indoor and outdoor aerosol. Insights and data presented will thus contribute to ongoing discus-
sions within the community of LIF users and will also stimulate discussions about needs for future
instrument improvements.

**2.    Materials and methods**
**2.1    Chemicals and materials**
Supplemental table S1 summarizes 19 polystyrene latex spheres (PSLs, 5 doped with fluorescent
dye) and 6 polystyrene divinylbenzene (PS-DVB) particles, which were purchased from Thermo
Fisher (Waltham, MA, USA), Bangs Laboratories Inc. (Fishers, IN, USA), Duke Scientific Corp.
(Palo Alto, CA, USA), and Polysciences Inc. (Warrington, PA, USA). A detailed study regarding
steady-state fluorescence properties of PSLs and PS-DVB particles used within this study can be





found in Könemann et al. (2018). Additionally, we analyzed particles comprised separately of seven

pure biofluorophores (tyrosine, tryptophan, NAD, riboflavin, chlorophyll *a* and *b*, and bacteriochlo-

rophyll) (Table S2) as well as one microorganism (*Saccharomyces cerevisiae;* baker's yeast, bought

at a local supermarket). Table S2 also includes reference particles used for asymmetry measurements,

namely iron oxide ($Fe_3O_4$), carbon nanotubes, and ammonium sulfate. Ultrapure water (MilliQ, 18

M$\Omega$) and $\geq$ 99.8 % ethanol (CAS Nr. 64-17-5, Carl Roth GmbH und Co. KG, Karlsruhe, Germany)

were used as solvents.

**2.2    Aerosolization of reference particles**

PSLs were aerosolized from aqueous suspensions with a portable aerosol generator (AG-100; DMT).

For both fluorescent and non-fluorescent PSLs, one drop of the suspension (or alternatively three

drops for 3 and 4 µm PSLs) was diluted into 10 ml ultrapure water in plastic medical nebulizers

(Allied Healthcare, St. Louis, MO, USA). The majority of water vapor from the aerosolization pro-

cess condenses inside the mixing chamber (~570 cm$^3$) of the aerosol generator. By using a tempera-

ture and relative humidity (RH) sensor (MSR 145 data logger, MSR Electronics GmbH, Seuzach,

Switzerland) monitoring the flow directly after the aerosol generator we measured RH values of

~33% (sample flow: 1.4 l/min, dilution: 5 l/min), ~39% (sample flow: 1.4 l/min, dilution: 4 l/min),

and ~54% (sample flow: 2.3 l/min, dilution: 2 l/min). Because of the low RH measured, we did not

use additional drying (e.g., diffusion dryer) to decrease humidity in the sample flow. Hence, the outlet

of the aerosol generator was directly connected to the SIBS inlet with ~30 cm of conductive tubing

($^1/_4$ inch). PSLs were measured for 1 min. Non-fluorescent 4.52 µm PSLs were measured for 2 min,

because of the low number concentrations due to poor aerosolization efficiency and gravitational

settling of larger particle sizes.

*S. cerevisiae* was analyzed using a method similar to the one stated above, with the exceptions

that the suspension was prepared with a spatula tip of material mixed into ultrapure water and that a

diffusion dryer (20 cm, 200 g silica) was added to remove excess water vapor. *S. cerevisiae* was

measured for 5 min. Chlorophyll *a*, *b*, and bacteriochlorophyll samples were diluted in 10 ml ethanol.

Between each measurement, the setup was cleaned by aerosolizing ultrapure water for 5 min.

PS-DVB particles and biofluorophores (Table S1 and S2) were aerosolized in a dry state. For this

purpose, air at a flowrate of ~0.6 l/min was sent through a HEPA filter into a 10 ml glass vial. A

small amount of each solid powder sample (~1 g) was placed inside the vial and entrained into the

particle-free airstream. Additionally, the sample was physically agitated by tapping the vial. The

outlet was connected with ~20 cm conductive tubing into the inlet of the SIBS. The tubing and glass



vial were cleaned after each measurement to prevent particle contaminations from previous measure-
ments. Each powder was sampled until cumulative number concentrations > 5000 particles were
reached.
In contrast to the monodisperse and spherical PSL standards, the biofluorophore aerosolization
process provided a polydisperse and morphologically heterogeneous particle distribution with sig-
nificant particle fractions at sizes < 1 µm. Therefore, we only used particles in a size range between
1 and 2 µm with sufficient fluorescence intensity values for subsequent data analysis. The only ex-
ceptions are the chlorophyll types, where a size range between 0.5 and 2 µm (chlorophyll *a* and *b*)
and 0.5 and 1 µm (bacteriochlorophyll) were used due to a less efficient particle aerosolization.
The fluorescent background of the SIBS was measured daily by firing the xenon lamps into the
optical chamber in the absence of particles (forced trigger mode). In this case, the diaphragm pump
was turned off and the inlet blocked to prevent particles reaching the optical chamber. One forced
trigger mode was performed per day with 100 xenon shots per min over a duration of 5 min. The
background signal (+ 1σ standard deviation (SD)) was subtracted from derived fluorescence emission
of each sample. Additionally, the background signal was reviewed periodically between each bio-
fluorophore measurement to verify that, e.g., optical components are not coated with residues from
previous measurements. A significant change in background signal was not observed between indi-
vidual measurements.
For particle asymmetry measurements, iron (II, III) oxide ($Fe_3O_4$), carbon nanotubes, and ammo-
nium sulfate were aerosolized in dry state, and 2 µm non-fluorescent PSLs and ultrapure water were
aerosolized with the aerosol generator method outlined above with SIBS integration times of 3 min
in all cases. Due to the broad distribution of asymmetry factor (AF) values for particles below 1 µm,
only the size fraction ≥ 1 µm was used for subsequent analyses. Furthermore, we observed that AF
bins between 0 and 1, and AF bin 100 tend to produce increased signal responses, especially for high
particle concentrations, for which they were discarded within the analyses. The origin of this effect
is unknown, but most likely related to detector noise.
For collection of particles for microscopy measurements, the sample flow was bypassed and led
through a custom-made particle impactor, which was connected to a mass flow controller (D-6321-
DR, Bronkhorst High-Tech B.V., Ruurlo, Netherlands) and a membrane pump (N816.1.2KN.18,
KNF, Freiburg, Germany). Particles were collected out of the sample flow onto glass cover slips (15
mm diameter) at a flow rate of 2 l/min over a duration of 1 min.

**2.3 Reference fluorescence spectra**
A Dual-FL fluorescence spectrometer (Horiba Instruments Incorporated, Kyoto, Japan) was used as
an offline reference instrument to validate the SIBS spectra. Aqualog V3.6 (Horiba) software was



used for data acquisition. The spectrometer was manufacturer-calibrated with NIST Fluorescence
Standard Reference Materials (SRMs 2940, 2941, 2942, and 2943). Aforementioned standard fluor-
ophores were analyzed using the SIBS excitation wavelengths at $\lambda_{ex}$ = 285 and 370 nm. The Dual-
FL[1] spectrometer uses a xenon arc lamp as excitation source and a CCD (charge-coupled device) as
emission detector, capable of detecting fluorescence emission between 250 and 800 nm. Unless oth-
erwise stated, a low detector gain setting (2.25 e$^-$ per count) and an emission resolution of 0.58 nm
was used for all measurements with the Dual-FL. Subsequently, we use the term "reference spectra"
for all measurements performed with the Dual-FL. In total, 100 individual spectra were recorded for
each sample and averaged spectra were analyzed in Igor Pro (Wavemetrics, Lake Oswego, Oregon
USA). Background measurements (solvent[2] in the absence of particles) were taken under the same
conditions as for sample measurements and subtracted from the emission signal. For direct compar-
ison to spectra recorded by the SIBS, reference spectra were re-binned by taking the sum of the
fluorescence intensity within the spectral bin width of each SIBS detection channel (Table 1).
For PSL measurements, 1.5 µl of each PSL stock solution was diluted in 3.5 ml ultrapure water
in a 10 x 10 x 40 mm UV quartz cuvette (Hellma Analytics, Müllheim, Germany) and constantly
stirred with a magnetic stirrer to avoid particle sedimentation during measurements. Chlorophyll *a*
and *b* and bacteriochlorophyll were handled equally, however concentrations were individually ad-
justed to prevent the detector from being saturated and to avoid self-quenching or inner filter effects
(Sinski and Exner, 2007). Concentrations were used as follows: chlorophyll *a*: 300 nmol/l, chloro-
phyll *b*: 1 µmol/l, and bacteriochlorophyll: 3 µmol/l. PSLs, chlorophyll *b*, and bacteriochlorophyll
measurements were performed with an integration time of 2 s. For chlorophyll *a* an integration time
of 1 s was used.
All other biofluorophores, *S. cerevisiae*, and PS-DVB particles were measured in dry state using
a front surface accessory (Horiba). The sample was placed into the surface holder and covered with
a synthetic fused silica window. To limit detector saturation from more highly fluorescent particle
types, the surface holder was placed at a 70° angle to the fluorescence detector for NAD and ribofla-
vin, 75° for tyrosine, 80° for *S. cerevisiae*, and 85° for tryptophan and PS-DVB particles and subse-
quently excited at $\lambda_{ex}$=285 and 370 nm. Emission resolution and detector gain settings were used as
for measurements of samples in solution, except for an integration time of 1 s for all dry samples.
Background measurements were performed as described above and subtracted from each sample.
Excitation-emission matrices (EEMs) were measured with the same samples as for single wavelength
measurements. EEMs were recorded at excitation wavelengths between $\lambda_{ex}$ = 240 and 800 nm (1 nm
increments) and an emission range between $\lambda_{em}$ = 247 and 829 nm (0.58 nm increments). Exposure

---

[1] Technical information taken from Dual-FL operation manual, rev. A, 30 NOV 2012; Horiba.
[2] Note that ≥ 99.8 % ethanol was used as solvent for chlorophyll *a*, *b*, and bacteriochlorophyll instead of ultrapure water.



times of 1 s were used, except for 2 µm green, 3 µm non-fluorescent PSLs (2 s), and NAD (0.5 s).
EEMs were analyzed using Igor Pro.

**2.4    Calibration lamps and spectral correction**
The relative responsivity of a fluorescence detector can vary substantially across its emission range
and, therefore, must be spectrally corrected as a function of emission wavelength (e.g., DeRose, 2007;
Lakowicz, 2004). For spectral correction it was important to choose: (i) light sources covering the
full spectral emission range of the SIBS, with temporal stability on the timescale of many months
and (ii) a calibrated and independent spectrometer to serve as spectral reference.

A deuterium-halogen lamp (DH-Mini; Ocean Optics, Largo, FL, USA) and a halogen projector

lamp (EHJ 24 V, 250 W; Ushio Inc., Tokyo, Japan) were used as calibration light sources. Both
lamps were connected to a 50 cm optical fiber (FT030, Thorlabs, Newton, NJ, USA) and vertically
fixed inside the optical chamber of the Dual-FL spectrometer. An aluminum mirror was attached to
the end fitting of the optical fiber, reflecting light in a 90° angle into the detector opening. The pro-
jector halogen lamp was allowed to warm up for 30 s before each measurement. For all power levels
(100, 150, 200, and 250 W), an integration time of 3 s was used. The DH-Mini was operational for
30 min before each measurement. Settings were used as for the projector halogen lamp, however,
due to the low emission a high detector gain setting (9 e$^-$ per count) was used with an integration time
of 25 s. As described in Sect. 2.3, 100 single measurements were taken and averaged (Fig. S1). For
the SIBS, both light sources were measured in the same way as for the reference spectra. Measure-
ments were performed with a detector amplification at 610 V (see Sect. 4.2). Background measure-
ments were taken as described in Sect. 2.2. Projector halogen lamp spectra (at all power levels) were
recorded for 3 min, the DH-Mini, due to its low emission intensity, for a duration of 5 min.

For the halogen projector lamp, averaged intensity values in each spectral bin were acquired at

each power level (150, 200, and 250 W). Spectra measured at 100 W were discarded due to the low
and unstable emission at wavelengths shorter than ~500 nm (Fig. S1). Reference spectra and spectra
recorded by the SIBS were normalized onto the SIBS detection channel 9 ($\lambda_{mean} = 528.0$ nm), which
is, theoretically, the detection channel with the highest responsivity (see Sect. 4.3). The individual
spectral correction factors were calculated by dividing the reference spectra by the spectra derived
from the SIBS. The final correction factors are a combination out of both light sources where the
detection channels 1-5 ($\lambda_{mean} = 302.2 - 415.6$ nm) include the correction factors for the DH-Mini and
the detection channels 6-16 ($\lambda_{mean} = 443.8 - 721.1$ nm) the correction factors for the halogen projector
lamp. At the intersection between channel 5 and 6, both corrections (DH-Mini, halogen) are in good
agreement ($\Delta_{correction} = 0.6$ in channel 6). For all particle measurements described in the following




298 sections, the background signal and raw sample spectra recorded by the SIBS were multiplied by

299 those correction factors.


301 **2.5 Microscopy of selected reference particles**

302 Bright field microscopy was conducted using an Eclipse Ti2 (Nikon, Tokyo, Japan) with a 60x im-

303 mersion oil objective lens and an additional optical zoom factor of 1.5, resulting in a 90x magnifica-

304 tion. Glass cover slips, used as collection substrates in the particle impactor (Sect. 2.2), were put onto

305 a specimen holder and fixed with tape. Images were recorded using a DS Qi2 monochrome micro-

306 scope camera with 16.25 megapixels and z-stacks of related images were created using the software

307 NIS-Elements AR (both Nikon).

309 **2.6 Ambient measurement setup and data analysis**

310 The SIBS was operated between the 5[th] of April to the 7[th] of May 2018 on the roof (fourth floor inside

311 a roof laboratory) of the Max Planck Institute for Chemistry in Mainz, Germany (49°59′28.2″N,

312 8°13′44.5″E) similar to measurements as described in Huffman et al. (2010) using a UV-APS. The

313 period between the 12[th] and 18[th] of April 2018 is described here to highlight the capability of the

314 SIBS to monitor ambient aerosol. Beside of the SIBS, four additional instruments (data not shown

315 within this study) were connected with ~20 cm conductive tubing ($\frac{1}{4}$ inch) to a sample airflow

316 splitter (Grimm Aerosol Technik GmbH & Co. KG, Ainring, Germany). The splitter was connected

317 to 1.5 m conductive tubing ($\frac{5}{8}$ inch), bent out of the window, and connected to 2.4 m stainless steel

318 tubing ($\frac{5}{8}$ inch, Dockweiler AG, Neustadt-Glewe, Germany) vertically installed. Between a TSP

319 head (total suspended particles, custom-made) and the stainless steel tubing, a diffusion dryer (1 m,

320 1 kg silica) was installed. Silica was exchanged every third to fourth day and periodic forced trigger

321 measurements were performed. The total flow was ~8.4 l/min.

322  For measurements presented here, only particles were included if they showed fluorescence emis-

323 sion in at least two consecutive spectral channels. This filter was applied to limit noise introduced

324 from measurement artifacts from a variety of sources and will need to be investigated in more detail.

325 The conservative analysis approach here suggests that the values reported are likely to be a lower

326 limit for fluorescent particle number and fraction. The observations are in line with previous meas-

327 urements, however, giving general support that the SIBS measurements are reasonable. Note that the

328 maximum repetition rate of the xenon lamps is 125 Hz, corresponding to maximum concentrations

329 of 20 particles per cm$^{-3}$ (see Sect. 3.3). Because ~50% of the total coarse particle number were excited

330 by xenon 1 and xenon 2, the fluorescent particle concentrations and fluorescent fractions are cor-

331 rected accordingly.



## 3. Design and components of the SIBS

The SIBS is based on the general optical design of the WIBS-4A (e.g., Foot et al., 2008; Healy et al., 2012; Hernandez et al., 2016; Kaye et al., 2005; Perring et al., 2015; Robinson et al., 2017; Savage et al., 2017; Stanley et al., 2011) with improvements based on a lower particle sizing limit, resolved fluorescence detection, and a broader emission range. The instrument provides information about size, particle asymmetry, and fluorescence properties for individual particles in real-time. The excitation wavelengths are optimized for the detection of the biological fluorophores tryptophan, NAD(P)H, and riboflavin. However, other fluorophores in PBAP will certainly fluoresce at these excitation wavelengths as many of them cluster in two spectral fluorescence "hotspots" as summarized in Pöhlker et al. (2012 and references therein)  and as shown for WIBS-4A measurements by Savage et al. (2017). Figure 1 shows an overview of excitation wavelengths and emission ranges of the UV-APS, WIBS-4A, WIBS-NEO, and SIBS for bioaerosol detection in relation to the spectral location of selected biofluorophores, such as tyrosine, tryptophan, NAD(P)H, riboflavin, and chlorophyll $b$. At $\lambda_{ex} = 285$ nm, the SIBS excites fluorophores in the "protein hotspot", at $\lambda_{ex} = 370$ nm fluorophores in the "flavin/coenzyme hotspot" (Pöhlker et al., 2012). In contrast to the UV-APS, the SIBS is able to detect fluorescence signals from chlorophyll due to the extended upper spectral range of detection (up to $\lambda_{em} = 721$ nm). Both the WIBS-4A and WIBS-NEO cover the spectral emission range for chlorophyll $b$, however, cannot provide resolved spectral information to separate it from other fluorophores. Table 2 summarizes and compares parameters and technical components of the SIBS, WIBS-4A, and WIBS-NEO. The individual components are described in detail in the subsequent sections.

### 3.1 Aerosol inlet and flow diagram

The design for the aerosol inlet of the SIBS is identical to the inlet of the WIBS-4A and WIBS-NEO. A detailed flow diagram is shown in Figure S2. Aerosol is drawn in via an internal pump as laminar air flow through a tapered delivery nozzle (Fig. S2a) where sheath (~2.2 l/min) and sample flow (~0.3 l/min) are separated.

### 3.2 Size and shape analysis

After passing the delivery nozzle, entrained particles traverses a 55 mW continuous-wave diode laser at $\lambda = 785$ nm (#2 in Fig. 2 and position #1 in Fig. S3). Unlike in the WIBS-4A and WIBS-NEO (635 nm diode laser), the triggering laser in the SIBS is in the near-infrared (IR) region (> 700 nm) and, therefore, outside the detectable emission range of the 16-channel photomultiplier tube (PMT) to avoid scattered light from the particle trigger laser being detected (see Fig. 1). The side and forward





scattered light is collected and used for subsequent measurements. Side scattered light is collected
by two concave mirrors, which are directed at 90° from the laser beam axis, and reflect the collected
light onto a dichroic beam splitter (#7 in Fig. 2). A PMT (H10720-20, Hamamatsu Photonics K.K.,
Japan) converts incoming light signals into electrical pulses, which are used for particle triggering
and sizing (#6 in Fig. 2). For the determination of the optical particle size, the SIBS uses a calculated
calibration curve according to the Lorenz-Mie Theory, assuming spherical and monodisperse PSLs
with a refractive index of 1.59 (Brandrup et al., 1989; Lorenz, 1890; Mie, 1908). Compared to aero-
dynamic sizing, which depends on particle morphology and density (e.g., Reid et al., 2003; Reponen
et al., 2001), the calculated optical diameter can vary significantly if the assumption of sphericity is
not fulfilled. In contrast, optical sizing is not as affected by differences in material density. The in-
strument operator must thus be aware of uncertainties in measured particle size due to, e.g., particle
morphology, spatial orientation of a particle when traversing the trigger laser or changing refractive
indices. In contrast to the WIBS-4A, the SIBS and WIBS-NEO detect a range of particle sizes be-
tween ~0.3 and 100 µm (nominal), achievable by using one PMT gain setting instead of switching
between a "Low Gain" and "High Gain" setting. Physical and technical details of this Gain-switching
method are patent pending and are not publicly available.

The forward-scattered light is measured by a quadrant PMT (#5 in Fig. 2) to detect the scatter
asymmetry for each particle (Kaye et al., 1991, 1996). A OG-515 long pass filter (Schott AG, Mainz,
Germany) prevents incoming light from the xenon flash lamps in a spectral range below 515 ± 6 nm
from reaching the Quadrant PMT. To calculate the AF, the root-mean-square variations for each
quadrant of the PMT of the forward-scattered light intensities are used (Gabey et al., 2010). The AF
broadly relates whether a particle is more spherical or fibril. Theoretically, for a perfectly spherical
particle, the AF would be 0, whereas an elongated particle would correspond to an AF of 100 (Kaye
et al., 1991). However, due to electrical and optical noise of the Quadrant PMT, the AF value of a
sphere is usually between ca. 2 and 6 (according to WIBS-4A service manual (DOC-0345 Rev A)).
Because the AF value depends on physical properties of optical components, the baseline for spher-
ical particles may shift even within identical instruments (Savage et al., 2017). For example, the study
by Toprak & Schnaiter (2013) reported an average AF value for spherical particles of 8 using a
WIBS-4A. In contrast, AF values shown by Foot et al. (2008) were, on average, below ~5 for
spherical particles measured with a WIBS-2s prototype.

### 3.3 Fluorescence excitation

Two xenon flash lamps (L9455-41, Hamamatsu) (#3 and #4 in Fig. 2) are used to induce fluores-
cence. They emit light pulses, which exhibit a broad excitation wavelength range of 185 to 2000 nm.
The light is optically filtered to obtain a defined excitation wavelength. Further information about



spectral properties of the xenon flash lamps can be found elsewhere (Specification sheet TLSZ1006E04, Hamamatsu, May 2015). Figure 3 displays relevant optical properties of the lamps and filters used within the SIBS, WIBS-4A, and WIBS-NEO. For the SIBS, a BrightLine® FF01-285/14-25 (Semrock Inc., Rochester, NY, USA) single-band bandpass filter is used with $\lambda_{mean} = 285$ nm and an effective excitation band[3] of 14 nm width is used for xenon 1. For xenon 2, the single-band bandpass filter BrightLine® FF01-370/36-25 (Semrock) is used with $\lambda_{mean} = 370$ nm and with an effective excitation band of 36 nm width. The only difference between all three instruments is that the WIBS-4A and WIBS-NEO use a different single-band bandpass filter for xenon 1 (Semrock, BrightLine® FF01-280/20-25; $\lambda_{mean} = 280$ nm; effective excitation band of 20 nm). The excitation light beam is focused on the sample flow within the optical cavity, resulting in a rectangular beam shape of ~5 mm by 2 mm. Xenon 1 is triggered when particles pass position 2 in Figure S3 and approximately 10 µs later xenon 2 is triggered as the particles move further to position 3 in Figure S3. After firing, the flash lamps need ~5 ms to recharge. During the recharge period, particles are counted and sized but no fluorescence information is recorded. The maximum repetition rate of the xenon lamps yields a measurable particle number concentration of ~2 x $10^4$ $l^{-1}$ (corresponding to 20 $cm^{-3}$).

Irradiance values from light sources becomes a crucial factor when interpreting derived fluorescence data of LIF instruments because the fluorescence intensity is directly proportional to the intensity of incident radiant power, described by the relationship:

$$F = \phi I_0 (1 - e^{-\varepsilon bc}) \tag{1}$$

$\phi$: quantum efficiency, $I_0$: intensity of incident light, $\varepsilon$: molar absorptivity, $b$: path length (cell), $c$: molar concentration (Guilbault, 1990).

To measure the irradiance of each xenon lamp after optical filtering, we used a thermal power head (S425C, Thorlabs), which was placed at a distance of 11.3 cm (focus length from xenon arc bow to sample flow intersection) from the xenon lamp measuring over a duration of 1 min at 10 xenon shots per s. By measuring new xenon lamps, we observed an average irradiance of 14.8 mW/cm$^2$ for xenon 1 and 9.6 mW/cm$^2$ for xenon 2, corresponding to ~154 % higher irradiance (spectrally integrated) from xenon 1. A second set of lamps, used intermittently for three years including several months of continuous ambient measurements and a lab study with high particle concentrations, exhibited average irradiance values of 10.8 mW/cm$^2$ (1σ SD 1.8 mW/cm$^2$) for xenon 1 and 4.9

---

[3] The effective excitation band is defined as "guaranteed minimum bandwidth" (GMBW), describing the spectral region a bandpass filter transmits light relative from the mean wavelength. For example, a GMBW of 14 nm means that light is transmitted in a 7 nm spectral range above and below the mean wavelength.



mW/cm$^2$ (1σ SD 1.9 mW/cm$^2$) for xenon 2, corresponding to ~220 % higher irradiance from xenon
1. Comparing the nominal, transmission-corrected irradiance data from the two xenon lamps pro-
vided by the lamp supplier (Fig. 3a and 3b, red dashed lines), an irradiance imbalance between xenon
1 and xenon 2 can be assumed for all three LIF instruments discussed here (SIBS, WIBS-4A, and
WIBS-Neo).
Results shown here are comparable to multiple WIBS studies (e.g., Hernandez et al., 2016;
Perring et al., 2015; Savage et al., 2017), where fluorescence emission intensities at $\lambda_{ex}$ = 280 nm
(xenon 1) also show a tendency to be higher than those at $\lambda_{ex}$ = 370 nm (xenon 2).

**3.4   Spectrally resolved fluorescence detection**
Fluorescence emission from excited particles is collected by two parabolic mirrors in the optical
cavity and delivered onto a custom-made dichroic beam splitter (Semrock, #7 in Fig. 2). The beam
splitter allows transmission of incoming light between ~300 and 710 nm, with an average transmis-
sion efficiency of 96%. At wavelengths shorter than 300 nm, the transmission decreases rapidly to <
20% at 275 nm. At the upper detection end of the SIBS ($\lambda_{mean}$ = 721 nm), the transmission efficiency
decreases to ~89%. The scattering light from the diode laser is reflected at a 90° angle onto the PMT
used for particle detection and sizing. At the excitation wavelength of 785 nm, the reflection effi-
ciency is stated at ~95% (Fig. S4).
After passing the dichroic beamsplitter, the photons are led into a grating polychromator (A
10766, Hamamatsu) (#8 in Fig. 2). A custom-made transmission grating (Hamamatsu) is used to
diffract incoming light within a nominal spectral range between 290.8 – 732.0 nm. In case of the
SIBS, a grating with 300 g/mm groove density and 400 nm blaze wavelength is used, resulting in a
nominal spectral width of 441.2 nm and a resolution of 28.03 nm/mm. After passing the transmission
grating, the diffracted light hits a 16-channel linear array multi-anode PMT (H12310-40, Hamama-
tsu) (#9 in Fig. 2) with defined mean wavelengths for each channel as shown in Table 1.
For each single particle detected, two spectra are recorded, at $\lambda_{ex}$ = 285 and 370 nm. The detect-
able band range of the PMT overlaps the excitation wavelength of xenon 2. Therefore, a notch optical
filter (Semrock) is placed between the optical chamber and the grating polychromator to prevent the
detector from being saturated. Incoming light at wavelengths shorter than 300 nm and from 362 to
377 nm is blocked from reaching the PMT resulting in a reduced spectral bin width for detection
channels 1, 3 and 4. The first three detection channels are omitted because their mean wavelengths
are below $\lambda_{ex}$ = 370 nm (see also Fig. 1). Accordingly, the emission spectra for xenon 2 excitation
begin at channel 4 ($\lambda_{mean}$ = 387.3 nm).
Technical data (xenon flash lamps, filters, dichroic beam splitter, PMT responsivity, and trans-
mission grating) described in the previous sections (3.3 and 3.4) were provided by Hamamatsu and



Semrock. Note that transmission/reflection efficiencies of the dichroic beam splitter, cathode radiant
sensitivity of the PMT, and diffraction efficiency data are modeled. Thus, individual components
may differ slightly from modeled values, even within the same production batch. Neither company
assumes data accuracy or provides warranty, either expressed or implied.
The SIBS was originally designed and marketed to record time-resolved fluorescence lifetime.
The fluorescence lifetime of most biofluorophores, serving as targets for bioaerosol detection, are
usually below 10 ns (e.g., Chorvat & Chorvatova, 2009; Herbrich, et al., 2012; O'Connor et al., 2014;
Richards-Kortum & Sevick-Muraca, 1996). However, by choosing xenon lamps as excitation source,
recording relevant fluorescence lifetimes in this ns range is hampered by the relatively long decay
time of the xenon lamp excitation pulse (~1.5 µs). In principle, fluorescence lifetime measurements
would be possible if the xenon lamps were replaced by appropriate laser excitation sources in the
SIBS optical design.

**3.5    Software components and data output**
The SIBS uses an internal computer (#10 in Fig. 2) with embedded LabView-based data acquisition
software allowing the user to control functions in real time and change multiple measurement param-
eters. As an example, the "Single Particle" tab out of the SIBS interface is shown in Figure S5. Here,
the user can define, e.g., the sizing limits of the SIBS (upper and lower threshold) and the minimum
size of a particle being excited by the xenon flash lamps. Furthermore, forced trigger measurements
can be performed while on this particular tab. Subsequently, the term "forced trigger measurement"
will be replaced by "background signal measurement". A local Wi-Fi network is installed so that the
SIBS can be monitored and controlled remotely. A removable hard drive is used for data storage.
Data is stored in a HDF5 format to minimize storage space and optimize data write speed. Resulting
raw data are processed in Igor Pro. As an example: by using a minimum sizing threshold of 500 nm,
the SIBS data output per day, operating in a relatively clean environment (~40 particles per cm$^{-3}$),
can span several hundreds of MB. In contrast, the data output can increase up to ~3 GB daily in
polluted areas (~680 particles per cm$^{-3}$). By lowering the minimum sizing threshold to 300 nm, the
data volume can exceed 10 GB per day when sampling in a moderately polluted environment (~180
particles per cm$^{-3}$).

**4.    Results and data validation**
**4.1    Validation of SIBS sizing**
To validate the optical sizing of the SIBS, twenty particle size standards were analyzed, covering a
broad size range from 0.3 to 20 µm in particle diameter. Overall, the particle size measurements from



the SIBS (optical diameter) show good agreement with the corresponding measurements of physical
diameter reported by PSL and PS-DVB manufacturers (Fig. 4). For the SIBS and WIBS-NEO, the
manufacturer states a nominal minimum size detection threshold of 300 nm. Figure 4 shows that a
linear response between optical particle size and physical particle size extends to at least 300 nm.
Smaller particles were not investigated. The upper size detection threshold is reported by the manu-
facturer to be nominally 100 µm. However, the upper limit was not investigated here due to the dif-
ficulty in aerosolizing particles larger than this. In most field applications, the upper particle size cut
is often far below this value due to unavoidable sedimentation losses of large particles in the inlet
system (e.g., Moran-Zuloaga et al., 2018.; Von der Weiden et al., 2009). Note that the size distribu-
tions of physical diameter for PS-DVB standards are broader compared to the PSL standards, as
reported by the manufacturer (Table S1). This also translates to broader distributions of optical di-
ameter measured by the SIBS than for the PSL particles. The 0.356 µm PSL sample was an outlier
with respect to the overall trend, showing an optical diameter of 0.54 µm. We suspect that this devi-
ation between physical and optical size can be explained by a poor quality of this particular PSL
sample lot rather than an instrumental issue, and so it was not included in the calculation of the trend
line (Fig. 4). Furthermore, the SIBS was shown to undersize the PSLs between 0.6 and 0.8 µm,
however, the overall trend exhibits a coefficient of determination of $r^2 > 0.99$.

As mentioned in Sect. 3.2, an important point regarding the SIBS and WIBS-NEO is that the size
calibration within the unit cannot be changed by the user, meaning that the PMT output voltages are
transformed directly to outputted physical diameter within the internal computer using a proprietary
calculation. It is still important, however, for the user to perform sizing calibration checks frequently
to verify and potentially post-correct particle sizing of all particle sizing instruments, including the
SIBS and WIBS-NEO.

## 4.2    Amplification of fluorescence detector

As with all optical detection techniques, adequate understanding of detection thresholds is an essen-
tial aspect of instrument characterization and use **(e.g., Jeys et al., 2007; Savage et al., 2017)**. Ap-
plication of appropriate voltage gain settings must be applied to the physical detection process so as
not to lose information about particles that cannot be recovered by post-processing of data. Yet par-
ticles in the natural atmosphere exhibit an extremely broad range of fluorescence intensities (many
orders of magnitude), arising from the breadth of quantum yields for fluorophores occurring in aer-
osols and from the steep increase of fluorescence emission intensity with particle size ($2^{nd}$ to $3^{rd}$
power) (e.g., Hill et al., 2015; Könemann et al., 2018; Sivaprakasam et al., 2011; Swanson &
Huffman, 2018). This range of fluorescence properties is generally broader than the dynamic range
of any single instrument, and so a UV-LIF instrument can be operated, e.g., to either: (i) apply a



higher detector gain to allow high sensitivity toward detecting weakly fluorescing particles, often
from rather small particles (< 1 µm), at the risk of losing fluorescence information for large or
strongly fluorescent particles due to detector saturation or (ii) apply a lower detector gain to prefer-
entially detect a wide range of more highly fluorescent particles, but at the risk of not detecting
weakly fluorescent or small particles.
Amplification voltage of the 16-channel PMT used in the SIBS can be adjusted between 500 and
1200 V. Each of the 16 detection channels can also be individually adjusted using digital gain settings
within the SIBS acquisition software. This channel-specific gain does not affect the amplification
process (e.g., the dynode cascade), but rather modifies the output signal of single detection channels
digitally. The digital gain is applied only after the signal collection process, and so cannot compensate
for a signal that is below the noise threshold or that saturates the detector. The digital gain was thus
left at the maximum gain level (255 arbitrary units (a.u.)) for all channels during particle measure-
ments discussed here.
To explore the influence of amplification voltage on particle detectability, 0.53 µm purple PSLs
were chosen to arbitrarily represent the lower limit of detectable fluorescence intensity. Using larger
(0.96 µm) particles comprised of the same purple fluorophore, Könemann et al. (2018) showed that
the particles were only narrowly detectable above the fluorescence threshold in each of the three
channels of a WIBS-4A (same unit as used in Savage et al., 2017) and so the smaller, 0.53 µm PSLs
were chosen here as a first proxy for the most weakly fluorescing particles we would expect to detect.
To improve the signal to noise ratio (SNR) for the lower fluorescence detection limit, the PMT am-
plification voltage was varied in seven steps between 500 and 1000 V (corresponding to a gain from
$10^3$ to $10^6$, specification sheet TPMO1060E02, Hamamatsu, June 2016) for purple PSLs and back-
ground signals (Fig. 5a). Whereas PSL spectra at a PMT amplification of 500 V were indistinguish-
able from the background signal (+ 1σ SD), spectra show a discernable peak at $\lambda_{mean} = 415.6$ nm
above 600 V. Subsequently, the SIBS was operated with a PMT amplification voltage of 610 V
corresponding to the lowest SNR threshold accepted (Fig. 5a, b). The detection of small biological
particles was tested by measuring the emission spectrum of *S. cerevisiae* as an example of a PBAP
(see also Pöhlker et al., 2012). On average, the size of intact *S. cerevisiae* particles range between ~2
– 10 µm (e.g., Pelling et al., 2004; Shaw et al., 1997). To test the ability of the SIBS to detect low
intensity emissions, we separately analyzed *S. cerevisiae* particles between 0.5 and 1 µm, which most
likely includes cell fragments caused by the aerosolization process (Fig. 5c). The tryptophan-like
emission, peaking in detection channel 2 ($\lambda_{mean} = 330.6$ nm) for $\lambda_{ex} = 285$ nm, reveals intensity values
below 100 a.u., which are comparable to fluorescence intensity values derived from 0.53 µm purple
PSLs (detection channel 5, $\lambda_{mean} = 415.6$ nm, Fig. 5d). These two tests for *S. cerevisiae* and 0.53 µm
purple PSLs confirmed the instruments ability to detect emission spectra from particles at least as



strongly fluorescent as these two test cases, leaving a wide range to detect larger and more intensely
fluorescing particles. By using a $3\sigma$ SD threshold, the fluorescence peak at $\lambda_{mean} = 415.6$ nm of 0.53
µm purple PSLs is still detectable but cannot be distinguished from the background signal at a $6\sigma$ SD
threshold anymore. Therefore, fluorescence intensity values at the lower detection limit should be
treated with care. Corrected spectra of both *S. cerevisiae* and 0.53 µm purple PSLs can be found in
the supplement (Fig. S6). By operating the SIBS at relatively low detector amplification, very weak
fluorescence, especially from small particles (< 1 µm) might not exceed the detection threshold dur-
ing field applications and would be missed. Further investigation will be necessary to choose ampli-
fication voltages appropriate for individual applications where smaller or otherwise weakly fluores-
cent particles might be particularly important. For all subsequent measurements discussed here, a
PMT amplification voltage of 610 V was used.
Saturation only occurred for 15 and 20 µm non-fluorescent PS-DVB particles. As highlighted in
Figure S7, the polystyrene/detergent signal (Könemann et al., 2018) at $\lambda_{ex} = 285$ nm for 10 µm PS-
DVB particles can be spectrally resolved (Fig. S7b), whereas the spectrum for 15 µm PS-DVB par-
ticles (Fig. S7e) is altered due to single particles (~10 % out of 400 particles) saturating the detector
(at 62383 a.u.). By comparing the defined lower detection end (Fig. 5) to the upper end (Fig. S7), a
quantitative difference of approximately three orders of magnitude can be estimated, indicating a
wide detectable range at the chosen amplification voltage setting.

### 4.3      Wavelength-dependent spectral correction of detector

The 16 cathodes of the PMT should be considered as independent detectors with wavelength depend-
ent, individual responsivity and amplification characteristics. In combination with physical properties
of technical components (e.g., excitation sources, optical filters, gratings), an instrumental-specific
spectral bias might result in incorrect or misleading spectral patterns if not corrected (e.g., DeRose,
2007; DeRose et al., 2007; Holbrook et al., 2006). To compensate for such potential instrumental
biases, we used a spectral correction approach as described in Sect. 2.4. The spectral correction fac-
tors are comparable to the theoretical responsivity of the PMT with the highest correction for chan-
nels 1-4 ($\lambda_{mean} = 302.2 - 387.3$ nm) and 14-16 ($\lambda_{mean} = 666.5 - 721.1$ nm) (Fig. 6). Channel 8 ($\lambda_{mean} =$
500.0 nm) shows the highest responsivity and channels 6 and 7 ($\lambda_{mean} = 443.8$ and 471.9 nm) exhibit
a noticeable lower responsivity than their adjacent channels (see also Sect. 4.4.1). The spectral cor-
rection shows several peaks (e.g., detector channels 3, 5, and 8) and dips (e.g., detector channels 4,
6, and 7) (Fig. 6), however, this pattern is due to gain variations for different channels and is not
noise.
It is important to note that the detector settings and spectral correction uniquely refer to the SIBS
unit as it was used for the current study. Due to technical and physical variability as stated above, it





is likely that the spectral correction required for other SIBS units would be somewhat different. Furthermore, the wavelength-dependent detector correction may change over time due to material fatigue or contaminations in the optical chamber affecting background signal measurements. Periodic surveillance and adjustments are therefore required, especially after measurements where the instrument was exposed to high particle concentrations or was operated during extreme weather or environmental conditions (e.g., temperature, humidity, vibration). For particle sizing verification, we recommend the use of 0.5, 1, and 3 µm non-fluorescent PSLs. Regarding a fluorescence response check, we recommend 2 µm green and 2 µm red PSLs for the validation of the spectral responsivity maximum and the upper (near-IR) detection range. To our knowledge, no fluorescent dyed PSLs are available to verify the response within the lower spectral detection range (UV) of the SIBS. However, the polystyrene signal of 3 µm non-fluorescent PSLs (Fig S7g, h, i, see also Könemann et al., 2018) represents a compromise between signal strength at $\lambda_{ex} = 285$ nm and aerosolization efficiency (compared to PSLs with larger sizes) for a spectral responsivity validation.

## 4.4 Fluorescence spectra of standards

### 4.4.1 PSL standards

The SIBS spectra for the four different PSL standards, covering an emission range from UV to near-IR, generally agree well with the corresponding reference spectra (Fig. 7). Each of the two excitation wavelengths probe separate fluorescent modes, which appear at approximately the same emission wavelength for a given PSL type (e.g., $\lambda_{em} = $ ~580 nm for red PSLs, Fig. 7j), as discussed by Könemann et al. (2018). Moreover, even the rather weak polystyrene and detergent fluorescence, systematically associated with PSL suspensions (Könemann et al., 2018), is resolved by the SIBS at $\lambda_{ex} = 285$ nm and $\lambda_{em} = $ ~300 nm (Fig. 7b, e, h, k). It is further noteworthy that emissions at $\lambda_{ex} = 285$ nm are generally higher than derived emissions at $\lambda_{ex} = 370$ nm (Fig. 7c, f, i, l), supporting the finding that a particle receives higher irradiance values from xenon 1 than from xenon 2 (see also Sect. 3.3).

As mentioned in Sect. 4.3, detection channels 6 and 7 require relatively large correction factors. For 2.07 µm purple PSLs (Fig. 7b, c), the SIBS spectra closely match the references spectra after correction. For the 2.1 µm blue PSLs (Fig. 7e, f), however, the corrected spectrum matches the reference spectrum well, except at detection channel 6 ($\lambda_{mean} = 443.8$ nm), where the SIBS spectrum is lower than the reference spectrum by approximately 50%. This effect was also observed for 1 µm blue PSLs (Thermo Fisher, B0100), doped with the same fluorophore (data not included in this study). The reason for this malfunction is unknown. Nevertheless, because this effect only occurs noticeably for highly fluorescent blue PSLs and NAD (see also Sect. 4.4.2), one explanation could be that the instrument-dependent dynode cascade (the electronic amplification stages) for this partic-



ular detection channel is suppressed, resulting in a lower amplification efficiency. In this case, rela-
tively low signals could be amplified correctly, whereas medium or high intensity emission could
only be amplified up to a certain level. The amplification threshold for detection channel 6 is, how-
ever, unknown and needs further verification.

**4.4.2   Biofluorophore standards**
Figure 8 and 9 highlight fluorescence spectra of different biofluorophores measured by the SIBS,
which correspond to related reference spectra (compare also Pöhlker et al., 2012), showing that amino
acids (fluorescence emission only at $\lambda_{ex}$ = 285 nm), co-enzymes and flavin compounds (fluorescence
emission at $\lambda_{ex}$ = 285 and 370 nm), and chlorophylls (fluorescence emission only at $\lambda_{ex}$ = 370 nm)
can be spectrally distinguished.

The uncorrected spectrum of tryptophan (Fig. S9) highlights the necessity of a spectral correction

to compensate for the low detector responsivity within the UV and near-IR bins. If the fluorescence
signal of tryptophan remains uncorrected, the spectra is shifted slightly to longer wavelengths (red-
shifted) due to the low responsivity of channel 2 in comparison to channel 3, resulting in misleading
spectral information. For NAD (Fig. 8h, i), fluorescence intensity values of channel 6 are lowered
due the suppressed amplification efficiency in this particular channel as described for blue PSLs
(Sect. 4.4.1).

All biofluorophores (except chlorophyll types) were aerosolized as dry powders (see Sect. 2.2)

to avoid fluorescence solvatochromism effects, means solvent-dependent spectral shifts relative to
the dry fluorophore state, which serves as a reference case here (e.g., Johnson et al., 1985). Solvato-
chromism of fluorophores in aqueous solution – the only atmospherically relevant case – typically
shifts fluorescence emissions to longer wavelengths due to the stabilized excited state caused by polar
solvents (Lakowicz, 2004). This spectral red-shift can be seen in Figure S10, where the peak maxi-
mum for NAD shows a difference of ~15 nm between a dry and water-solved state, whereas ribofla-
vin reveals an even higher shift of ~37 nm. Here, solvatochromism serves as an example for fluores-
cence spectra that vary substantially as a function of the fluorophore's microenvironments (e.g., sol-
vent polarity, pH, temperature).

Each of the three types of chlorophyll exhibit the weakest emission of all biofluorophores meas-

ured within this study, however the SIBS was able to detect the fluorescence signal at $\lambda_{ex}$ = 370 nm
for all three (Fig. 9). The spectral difference between chlorophyll *a* and *b* is only minor at $\lambda_{ex}$ = 370
nm ($\Delta\lambda$ = 8.3 nm) for which the spectral resolution of the SIBS is not capable of distinguishing be-
tween both types (Fig. 9a, b, c, d and Fig. S11) (e.g., French et al., 1956; Welschmeyer, 1994). Nev-
ertheless, the SIBS shows the ability to distinguish between chlorophyll *a* and *b*, and bacteriochloro-
phyll due to the red-shift in the bacteriochlorophyll spectrum ($\Delta\lambda$ = 28.5 nm at $\lambda_{ex}$ = 370 nm, between



chlorophyll *a* and bacteriochlorophyll). This may provide a further discrimination level regarding
algae, plant residues, and cyanobacteria. Bacteriochlorophyll also shows a second and even stronger
emission peak at $\lambda_{ex}$ = 370 nm ($\lambda_{em}$ = ~800 nm) that could help further distinguish it from chlorophyll
*a* and *b*, but the SIBS spectrometer cannot currently detect this far into the IR (e.g., Rijgersberg et
al., 1980; Van Grondelle et al., 1983).

Overall, fluorescence emissions recorded by the SIBS are in good agreement with measured ref-

erence spectra. However, care must be taken as to the interpretation of fluorescence emissions cov-
ering broad spectral ranges, which span regimes with large differences between individual correction
factors (e.g., channel 15 ($\lambda_{mean}$ = 693.9 nm, Fig.7l) and channel 2 ($\lambda_{mean}$ = 330.6 nm, Fig.8k). For the
SIBS, namely the first two UV detection channels and the last two near-IR channels have to be treated
with care. Further investigations are required for a careful assessment of how the spectral correction
can be applied properly onto fluorescent and non-fluorescent atmospheric particles.

### 4.5    Particle asymmetry measurements

The AF of spherical particles such as PSLs (Fig. 10a, b) and ultrapure water droplets is approximately
10 (Table 3), which is slightly higher than reported values for spherical particles by, e.g.,  Savage et
al. (2017) (AF= ~5) or Toprak & Schnaiter (2013) (AF= ~8) using a WIBS. It is noteworthy that the
AF of water droplets increases slightly with increasing droplet size and, therefore, contributes to the
mean value (Fig. S13). This effect is most likely based on a decreasing surface tension with increas-
ing droplet size for which the droplet morphology is changed to a more oval shape within the sample
flow. A similar effect regarding a potential droplet deformation using an Airborne Particle Classifier
(APC) was observed by Kaye et al., (1991). Even if the morphology of ammonium sulfate (crystal-
line, Fig. 10d) and $Fe_3O_4$ (irregular clusters, Fig. 10f) is diverse, their differences in AFs is only
minor (~13 and 14, Table 3), indicating that most naturally occurring aerosols (e.g., sea salt, soot,
various bacterial and fungal clusters) will occur in a AF regime between ~10 and 20. Only rod-shaped
carbon nanotubes (110-170 nm diameter, 5-9 µm length) show increased AFs with a mean value at
~22 (Table 3) at which also, e.g., bacteria would occur (Fig. 10h). No particles observed exhibited
average AF values >25, as would have been expected for, e.g., carbon nanotubes. Because the range
of AF values for homogenous particles is relatively broad and the differences between morphologi-
cally diverse particle types is only minor (Table 3), the question can be raised to what extent particles
could be distinguished based on the AF under ambient conditions. As also discussed by Savage et al.
(2017), the AF values reported by SIBS and WIBS units should be treated with extreme care.

The validation of asymmetry measurements is challenging due to unavoidable particle and aero-

solization effects (e.g., particle agglomeration and spatial orientation within the sample flow) and the





lack of standardized procedures for AF calibrations. Measurements performed in this study do, there-
fore, only serve as a rough AF assignment. Moreover, even if both the SIBS and WIBS use the same
technical components for defining AFs, a direct intercomparison cannot be applied due to technical
variability (e.g., PMT related signal-to-noise ratio or the alignment of optical components). Addi-
tionally, it is currently unknown in how far the 785 nm diode laser of the SIBS affect asymmetry
measurements compared to the WIBS using a 635 nm diode laser.

**4.6    Initial ambient measurements**

Several weeks of initial ambient SIBS measurements have been conducted on the roof of the Max
Planck Institute for Chemistry in Mainz, Germany. At the same location, Huffman et al. (2010) con-
ducted one of the first ambient UV-APS studies. Moreover, Toprak & Schnaiter (2013) conducted a
WIBS-4A study at a comparable site in central Germany. The aim of this brief ambient section is to
validate that the SIBS-derived key aerosol and fluorescence data are consistent with the aforemen-
tioned studies. We found a good agreement between the coarse mode ($\geq 1\,\mu m$) number concentrations
($N_{\text{T,c}}$) of the SIBS ($N_{\text{T,c}}$ ranging from 0.19 to 1.24 cm$^{-3}$, with a mean of 0.59 cm$^{-3}$), the UV-APS (mean
$N_{\text{T,c}}$: 1.05 cm$^{-3}$ (Huffman et al., 2010)), and the WIBS-4A (mean $N_{\text{T,c}}$: 0.58 cm$^{-3}$ (Toprak and
Schnaiter, 2013)) (Fig.11a). Furthermore, good agreement was found between coarse mode fluores-
cent number concentrations ($N_{\text{F,c}}$) of the SIBS with a 3σ SD threshold (mean $N_{\text{F,c (3σ)}}$: 0.019 cm$^{-3}$),
the UV-APS (mean $N_{\text{F,c}}$: 0.027 cm$^{-3}$ (Huffman et al., 2010)), and the WIBS-4A with a 3σ SD thresh-
old (mean $N_{\text{F,c (3σ)}}$: 0.031 cm$^{-3}$ (Toprak and Schnaiter, 2013)) (Fig.11a). Similarly, the fraction of
fluorescent particles in the coarse mode ($N_{\text{F,c}}/N_{\text{T,c}}$) compares well between SIBS with a 3σ SD thresh-
old (mean $N_{\text{F,c(3σ)}}/N_{\text{T,c}}$: 4.2 %), the UV-APS (mean $N_{\text{F,c}}/N_{\text{T,c}}$: 3.9 % (Huffman et al., 2010)), and the
WIBS-4A with a 3σ SD threshold (mean $N_{\text{F,c(3σ)}}/N_{\text{T,c}}$: 7.3 % (Toprak and Schnaiter, 2013)) (Fig.11b).
Expectedly, a 1σ SD threshold gives much higher fluorescent fractions of 39.2 %, whereas a 6σ SD
threshold corresponds with much lower fluorescent fractions of 1% (Fig.11b). Note that no prefect
match between our results and the studies by Huffman et al. (2010), and Toprak & Schnaiter (2013)
can be expected, since the measurements took place with different sampling setups and during dif-
ferent seasons. Furthermore, the spectrally resolved SIBS data makes the definition of fluorescent
fraction more complex than for UV-APS and WIBS data (see Sect. 2.6). However, the overall good
agreement confirms that the SIBS produces reasonable results in an ambient setting. Further, the
single particle fluorescence spectra are reasonable with respect to typical biofluorophore emissions
(Pöhlker et al., 2012). Exemplary spectra ($\lambda_{\text{ex}} = 285$ and 370 nm) of ambient single particles can be
found in the supplement (Fig.S14). An in-depth analysis of extended SIBS ambient datasets is subject
of ongoing work.



## 5.    Summary and conclusions

Real-time analysis of atmospheric bioaerosols using commercial LIF instruments has largely been restricted to data recorded in only 1-3 spectrally integrated emission channels, limiting the interpretation of fluorescence information. Instruments that can record resolved fluorescence spectra over a broad range of emission wavelengths may thus be required to further improve the applicability of LIF instrumentation to ambient PBAP detection. Introduced here is the SIBS as a new aerosol fluorescence detector, which provides resolved fluorescence spectra ($\lambda_{mean} = 302 - 721$ nm) from each of two excitation wavelengths ($\lambda_{ex} = 285$ and 370 nm) for single particles. The current study introduces the SIBS by presenting and experimentally validating its key functionalities. This work critically assesses the strengths and limitations of the SIBS with respect to the growing interest in real-time bioaerosol quantification and classification. It should be noted that the study is an independent evaluation that was not conducted or endorsed by the manufacturer. Overall, this work confirms a precise particle sizing between 300 nm and 20 µm and the particle discrimination ability based on spectrally resolved fluorescence information of several standard compounds.

The SIBS was operated at a low PMT detector amplification setting (610 V) to retain capacity to detect large or brightly fluorescent particles. It was confirmed, however, that even weak fluorescence signals from 0.53 µm purple PSLs and from small *S. cerevisiae* fragments (0.5 - 1 µm) can be clearly distinguished from the background signal. Saturation events were only observed for the polystyrene/detergent signal from relatively large 15 and 20 µm PS-DVB particles. Nevertheless, the fluorescence intensity detection threshold is highly instrument-dependent due to the complex interaction of single technical components across individual instruments. For example, xenon 1 exhibited ~154 % higher irradiance than xenon 2 (both new lamps) due to differences in the properties of xenon emission and the optical filters used. For used xenon lamps (> 4000 hours of use), an even higher difference of ~220 % was observed. Thus, a defined fluorescence detection threshold will most likely change over time due to, e.g., material fatigue. These observations are valid not only for the SIBS, but also for the WIBS-4A and WIBS-NEO and lead to important implications for interpretation of particle data. In particular, a particle that exhibits measurable fluorescence in WIBS channel FL1, but only weak fluorescence in channel FL3 could be assigned as an "A-type" particle in one instrument but an "AC-type" particle in an instrument with slightly stronger xenon 2 irradiance. These differences in classification can be extremely important to interpretation of ambient data (e.g., Savage et al., 2017; Perring et al. 2015).

The PMT used in the SIBS shows a wavelength-dependent sensitivity distribution along all 16 detection channels. To compensate for this characteristic and to be able to use the broadest possible fluorescence emission range, the measured emission spectra were corrected with respect to reference spectra acquired from deuterium and halogen lamps. A spectral correction over a broad emission





range also introduces drawbacks, however, that LIF-instrument users should keep in mind while interpreting derived fluorescence information. In particular, the first two (UV) and the last two (near-IR) detection channels should be treated with care, because they have larger correction factors compared to adjacent channels. Ultimately, the correction factor and amplification voltages applied to the detector will be experiment-specific and will need to be investigated with respect to individual experimental aims.

Fluorescence spectra of fluorescent PSLs, amino acids, co-enzymes, and flavins measured by the SIBS agree well with corresponding spectra recorded with an offline reference spectrometer. Thus, the SIBS was shown to be capable of clearly distinguishing between different particle types based on resolved fluorescence information. Furthermore, the extended fluorescence emission range ($\lambda_{em} = > 700$ nm) enables the SIBS also to distinguish chlorophyll *a* and *b* from bacteriochlorophyll, potentially opening new possibilities for the detection of, e.g., algae, plant residues, and cyanobacteria.

Particle asymmetry measurements revealed that spherical PSLs have an AF of 9.9 (± 3.6), whereas other materials (ammonium sulfate, $Fe_3O_4$, and carbon nanotubes) show AF values of 13.1 (± 8.1), 14.4 (± 7.4), and 21.6 (± 12.7), respectively. Because differences of measured AF value between morphologically diverse particle types are small and within the ranges of uncertainty for the measurement of a given set of particles, it is questionable how well particles can be distinguished based on the AF by the quadrant PMT as presently measured. Users of SIBS and WIBS instruments should apply extreme care if using AF data. It is also likely that different instrument units may have very different AF responses with respect to this measurement. At a minimum, each individual unit needs to be rigorously calibrated to known particle types to determine if AF values are sufficiently different (e.g., separated by several standard deviations) to justify scientific conclusions based on the metric.

Exemplary ambient data, measured between the 12th and 18th of April 2018 on the roof of the Max Planck Institute for Chemistry in Mainz (Germany), are consistent with LIF measurement data using a UV-APS (Huffman et al., 2010) and a WIBS-4A (Toprak and Schnaiter, 2013). Total coarse particle number concentrations revealed a mean value of 0.59 cm$^{-3}$ (1.05 cm$^{-3}$ (Huffman et al., 2010); 0.58 cm$^{-3}$ (Toprak and Schnaiter, 2013)) of which ~4.2% are considered to be fluorescent using a 3σ SD threshold (3.9% (Huffman et al., 2010); 7.3% (Toprak and Schnaiter, 2013)), including only particles that show fluorescence emission in, at least, two adjoining detection channels. Using a 1σ and 6σ SD threshold results in fluorescent fractions of 39.2% and 1% respectively. However, the applicability of different threshold strategies for the SIBS is currently under investigation and needs further verifications.

The results suggest that the SIBS has the ability to increase the selectivity of detection of fluorescent biological and non-biological particles by use of two excitation wavelengths and 16-channel



resolved fluorescence information in combination with a broad detectable emission range. The ap-
plicability of described methods onto ambient datasets is currently under investigation. Data shown
here and the detailed insights of technical components used in the SIBS will be broadly beneficial
for users of LIF instruments providing resolved fluorescence information, but also for users of vari-
ous generations of WIBS and other LIF instruments widely applied within the bioaerosol community.

**6.    Data availability**
The data of the key results presented here can be provided upon request. For specific data requests,
please refer to the corresponding authors.

**Acknowledgements**
This work was supported by the Max Planck Society (MPG) and the Max Planck Graduate Center
with the Johannes Gutenberg-University Mainz (MPGC). Financial support for Nicole Savage was
provided by the Phillipson Graduate Fellowship from the University of Denver. We thank Maria
Praß, Jan-David Förster, Meinrat O. Andreae, Peter Hoor, Viviane Després, Benjamin Swanson,
Jorge Saturno, Bruna Holanda, Florian Ditas, Daniel Moran-Zuloaga, Björn Nillius, Jing Ming,
Gavin McMeeking, Gary Granger, Alexis Attwood, Greg Kok, Robert MacAllister, John Walker,
Matt Mahin, Matt Freer, Uwe Kuhn, Minghui Zhang, Petya Yordanova, Naama Lang-Yona, and
members of the Mainz Bioaerosol Laboratory (MBAL) for their support and stimulating discussions.



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





**Appendix A:** List of acronyms and symbols.

| Acronym/Symbol | Description |
| --- | --- |
| AF | Asymmetry factor |
| APC | Airborne Particle Classifier |
| CCD | Charge-coupled device |
| DMT | Droplet Measurement Technologies |
| EEM | Excitation-emission matrix |
| EM | Emission |
| EX | Excitation |
| IR | Infrared |
| LIF | Light-induced fluorescence |
| $N$ | Particle number concentration ($cm^{-3}$) |
| $N_{T,c}$ | $N$ of total coarse particles (1-20 µm) |
| $N_{F,c(n\sigma)}$ | $N$ of fluorescent coarse particles (1-20 µm) at 1, 3, or 6σ |
| NAD | Nicotinamide adenine dinucleotide |
| NAD(P)H | Nicotinamide adenine dinucleotide and nicotinamide adenine dinucleotide phosphate |
| NIST | National institute of standards and technology |
| PBAP | Primary biological aerosol particles |
| PMT | Photomultiplier tube |
| PAH | Polycyclic aromatic hydrocarbons |
| PSL | Polystyrene latex sphere |
| PS-DVB | Polystyrene-divinylbenzene |
| SD | Standard deviation |
| SIBS | Spectral intensity bioaerosol sensor |
| SNR | Signal to noise ratio |
| TSP | Total suspended particles |
| UV | Ultraviolet |
| UV-APS | Ultraviolet aerodynamic particle sizer |
| Vis | Visible light |
| WIBS | Wideband integrated bioaerosol sensor |



**Table 1.** Lower, mean, and upper wavelength at each PMT detection channel. Nominal data according to manufacturer Hamamatsu.

| Channel | $\lambda_{lower}$ (nm) | $\lambda_{mean}$ (nm) | $\lambda_{upper}$ (nm) |
|---------|------------------------|-----------------------|------------------------|
| 1 | 298.2 | 302.2 | 316.2 |
| 2 | 316.6 | 330.6 | 344.6 |
| 3 | 345.0 | 359.0 | 362.5 |
| 4 | 377.5 | 387.3 | 401.3 |
| 5 | 401.5 | 415.6 | 429.7 |
| 6 | 429.8 | 443.8 | 457.8 |
| 7 | 457.9 | 471.9 | 485.9 |
| 8 | 486.0 | 500.0 | 514.0 |
| 9 | 514.0 | 528.0 | 542.0 |
| 10 | 541.9 | 555.9 | 569.9 |
| 11 | 569.7 | 583.7 | 597.7 |
| 12 | 597.4 | 611.4 | 625.4 |
| 13 | 625.0 | 639.0 | 653.0 |
| 14 | 652.8 | 666.5 | 680.2 |
| 15 | 679.9 | 693.9 | 707.9 |
| 16 | 707.1 | 721.1 | 735.1 |



**Table 2.** Parameters and technical components of the SIBS in comparison to the WIBS-NEO and WIBS-4A. Data are taken from manufacturer information.

| | **SIBS** | **WIBS-NEO** | **WIBS-4A** |
|---|---|---|---|
| **Measured parameters** | Particle size<br>Asymmetry Factor<br>Fluorescence spectra | Particle size<br>Asymmetry Factor<br>Integrated fluorescence in 3 channels | Particle size<br>Asymmetry Factor<br>Integrated fluorescence in 3 channels |
| **Particle size range** | ~0.3 – 100 µm | ~0.3 – 100 µm | ~0.5 – 20 µm |
| **Maximum concentration** | ~2 x $10^4$ particles/L | ~2 x $10^4$ particles/L | ~2 x $10^4$ particles/L |
| **Fluorescence excitation** | $\lambda_{ex}$= 285 and $\lambda_{ex}$= 370 nm | $\lambda_{ex}$= 280 and $\lambda_{ex}$= 370 nm | $\lambda_{ex}$= 280 and $\lambda_{ex}$= 370 nm |
| **Fluorescence emission** | $\lambda_{mean}$= 302 – 721 nm<br>(16-channel PMT) | $\lambda_{em}$= 310-400 nm and<br>$\lambda_{em}$= 420-650 nm | $\lambda_{em}$= 310-400 nm and<br>$\lambda_{em}$= 420-650 nm |
| **Flow rate** | Sample flow:~0.3 l/min<br>Sheath flow:~2.2 l/min<br>(re-circulating) | Sample flow:~0.3 l/min<br>Sheath flow:~2.2 l/min<br>(re-circulating) | Sample flow:~0.3 l/min<br>Sheath flow:~2.2 l/min<br>(re-circulating) |
| **Laser** | 785 nm diode laser, 55 mW | 635 nm diode laser, 15 mW | 635 nm diode laser, 12 mW |
| **Pump** | Diaphragm pump | Diaphragm pump | Diaphragm pump |
| **Power requirements** | 200 W, 90 - 230 VAC | 150 W, 90 - 230 VAC | 150 W, 90 - 230 VAC |
| **Weight (kg)** | 20.1 | 12.5 | 13.6 |
| **Dimension W x L x H (cm)** | 42.5 x 61.5 x 23.5 | 45.1 x 36.2 x 24.1 | 30.4 x 38.2 x 17.1 |





**Table 3.** Asymmetry factor (AF) values for reference particles. Values are based on the mean of a

Gaussian fit applied onto each particle histogram (see also Fig. 10), including 1σ SD.

|  | AF |
| --- | --- |
| **2 μm non-fluorescent PSLs** | 9.9 ± 3.6 |
| **Ultrapure water** | 11.9 ± 2.9 |
| **Ammonium sulfate** | 13.1 ± 8.1 |
| **Fe₃O₄** | 14.4 ± 7.4 |
| **Carbon nanotubes** | 21.6 ± 12.7 |



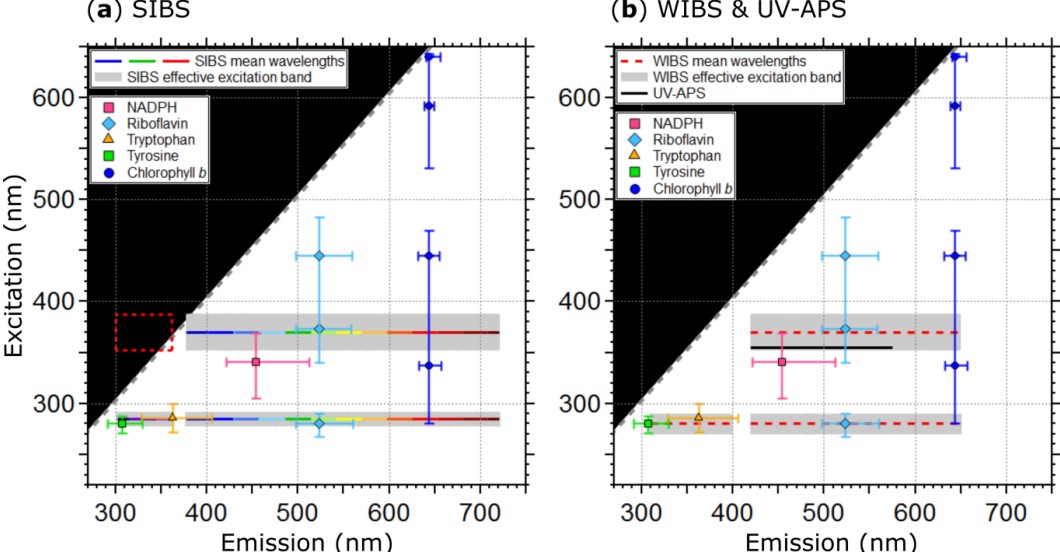

**Figure 1.** Optical design and overview of excitation and emission specifications of the LIF instruments UV-APS, WIBS, and the SIBS with spectral locations of the autofluorescence modes of the biofluorophores tyrosine, tryptophan, NAD(P)H, riboflavin, and chlorophyll *b* (as examples). Here the term WIBS includes the WIBS-4A and WIBS-NEO, because both instruments use the same optical components. Spectral properties of the emission bands of LIF instruments are illustrated as horizontal lines. The color-coded bars in (**a**) illustrate the spectrally resolved fluorescence detection of the two excitation wavelengths ($\lambda_{ex}$ = 285 and 370 nm) by the SIBS. The "blind spot" (white notch) at $\lambda_{ex}$ = 285 nm between $\lambda_{em}$ = 362 - 377 nm (**a**) originates from a notch optical filter, used to block incident light from the excitation sources. Grey dashed lines show the 1$^{st}$ order elastic scattering. At $\lambda_{ex}$ = 370 nm, the detection range of the SIBS includes the spectral range where $\lambda_{em} < \lambda_{ex}$, for which fluorescence is not defined and so data within the red dashed rectangle is omitted (**a**). Grey bars indicate the effective excitation bands of optical filters used for the WIBS and SIBS (see also Sect. 3.3 and Fig. 3). The effective excitation bands in the WIBS and SIBS occur in a spectral range spanning several nanometers (up to 36 nm), in contrast to the UV-APS (black line, **b**), which uses a laser source with a defined excitation (Figure adapted from Pöhlker et al., 2012).





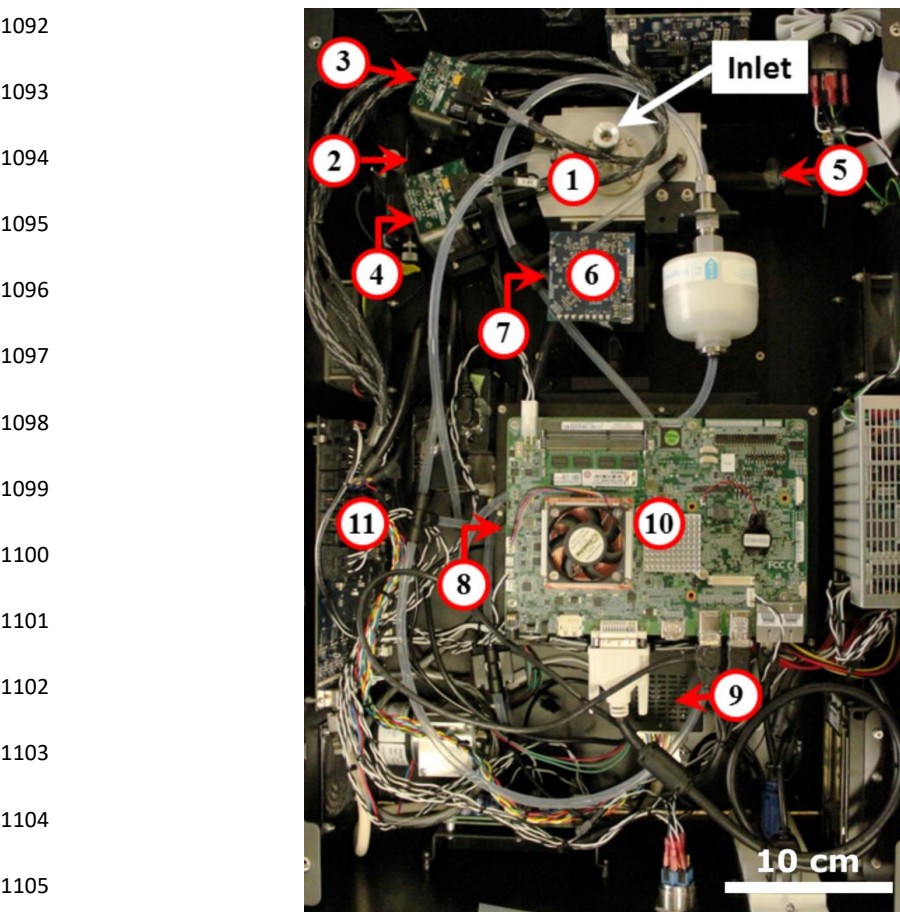

**Figure 2.** Technical components within SIBS body. (1) Optical cavity. (2) Continuous wave diode laser used for particle detection and sizing. (3) and (4) Xenon light sources. (5) Quadrant PMT used for the determination of particle asymmetry. (6) PMT used for particle detection and sizing. (7) Dichroic beamsplitter separates side-scattered light (particle sizing) and fluorescence emission (not visible; below component (6)). (8) Grating polychromator (below component (10)). (9) 16-channel PMT used for detection of fluorescence. (10) Embedded computer unit. (11) Control-board.





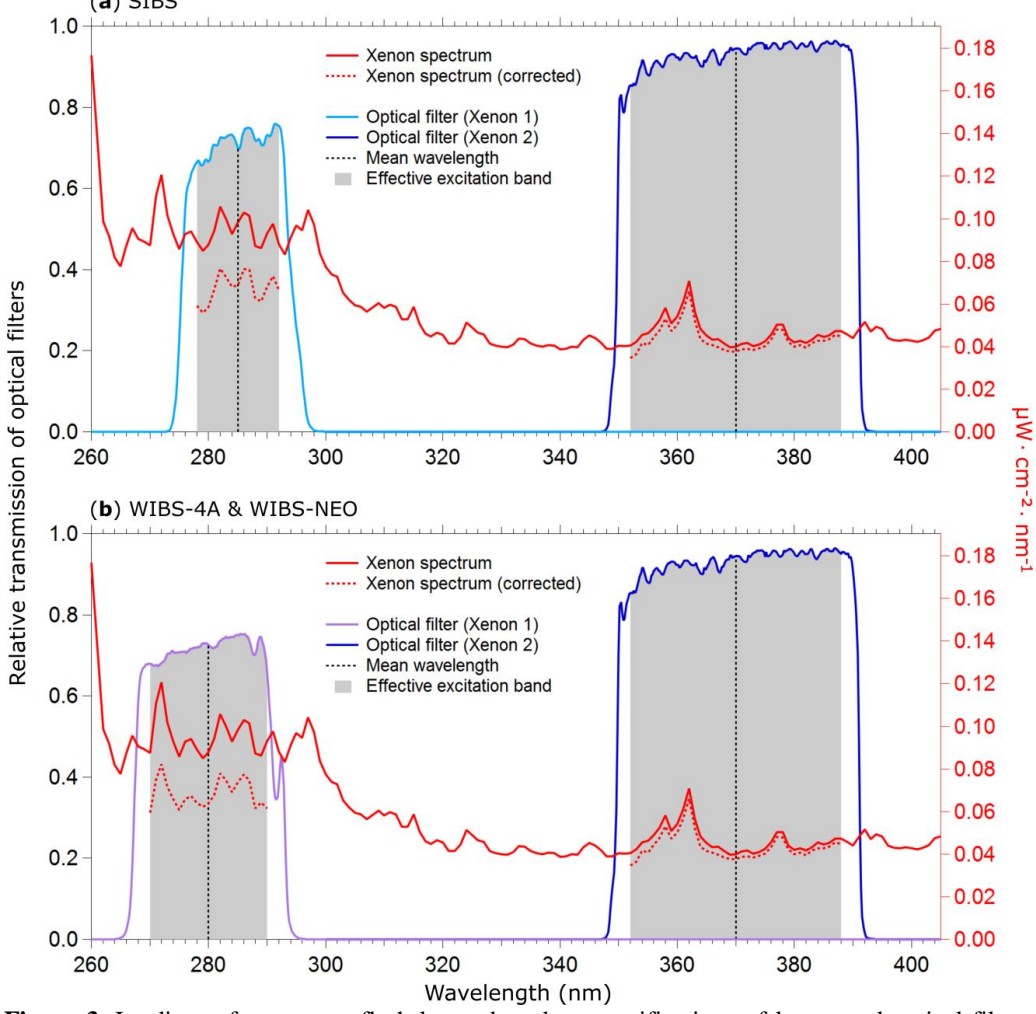

**Figure 3.** Irradiance from xenon flash lamps based on specifications of lamps and optical filters. Purple and blue lines show optical transmission of filters (left axes) applied to select excitation wavelength. Gray bands indicate where filter transmit light relative from the mean wavelength. Red lines show theoretical irradiance values of the xenon flash lamp (right axes): solid line (raw output), dashed line (relative output after filtering). Relative output shown as raw output multiplied by effective excitation band of the bandpass filters used in the: (**a**) SIBS ($\Delta\lambda_{ex\,(Xenon1)}$ = ~14 nm; $\Delta\lambda_{ex\,(Xenon2)}$ = ~36 nm), and (**b**) WIBS-4A and WIBS-NEO ($\Delta\lambda_{ex\,(Xenon1)}$ = ~20 nm; $\Delta\lambda_{ex\,(Xenon2)}$ = ~36 nm). Xenon lamp operating conditions: 600 V main voltage, 0.22 µF main capacitance, 126 Hz repetition rate, 500 mm distance. (Data courtesy: Xenon flash lamps / Hamamatsu; Single-band bandpass filters / Semrock).

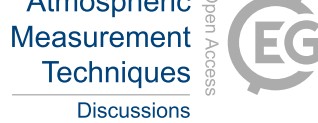

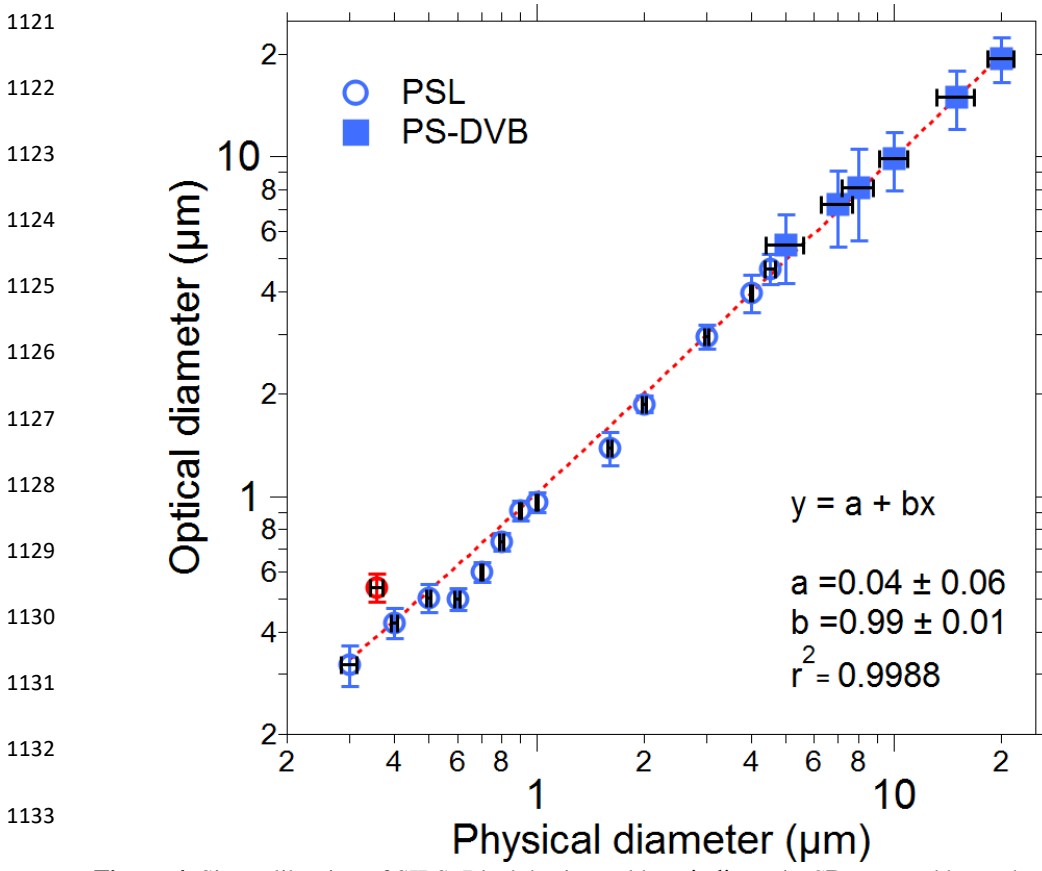

**Figure 4.** Size calibration of SIBS. Black horizontal bars indicate 1σ SD as stated by each manufacturer (Table S1). Optical diameter values and related 1σ SD are based on a Gaussian fit, which was used to average size distributions of several thousand homogeneous particles for each measurement. The linear fit (red dashed line) excludes the 0.356 µm PSL sample (red marker), an outlier potentially caused by a poor quality PSL batch. Only non-fluorescent particle standards were used for determining the sizing accuracy.



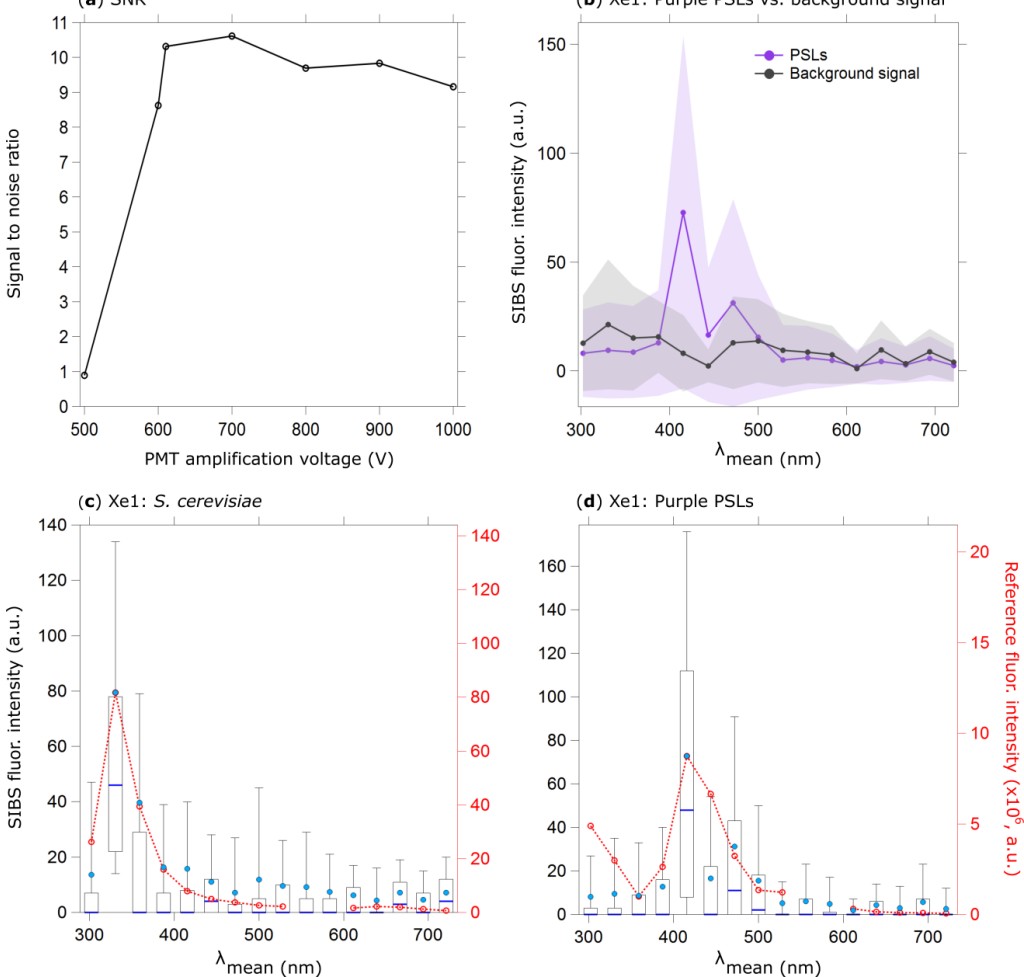

**Figure 5.** SIBS signal to noise ratio (SNR) in (**a**): emission of 0.53 µm purple PSLs (5260 particles, background signal + 1σ SD subtraction) divided by background signal at different PMT amplification voltages (both at Xe1, channel 5, averaged, and uncorrected). Background signal measured over 5 min. In (**b**), fluorescence emission in contrast to background signal at a PMT amplification voltage of 610 V are shown (same parameters as in (**a**)). Shaded area: 1σ SD. Fluorescence intensity values are shown in arbitrary units. Fluorescence emission spectra of (**c**) *S. cerevisiae* (yeast; 2048 particles, 0.5 – 1 µm) and (**d**) PSLs (as in (**b**)). Red dashed lines and markers (right axes) show averaged and re-binned reference spectra. Box and whisker plots (left axes) show SIBS spectra: median (blue line), mean (circle), boxes 75 and 25 percentile, whiskers 90 and 10 percentile. Data coinciding with 1st or 2nd order elastic scattering were removed from reference spectra.





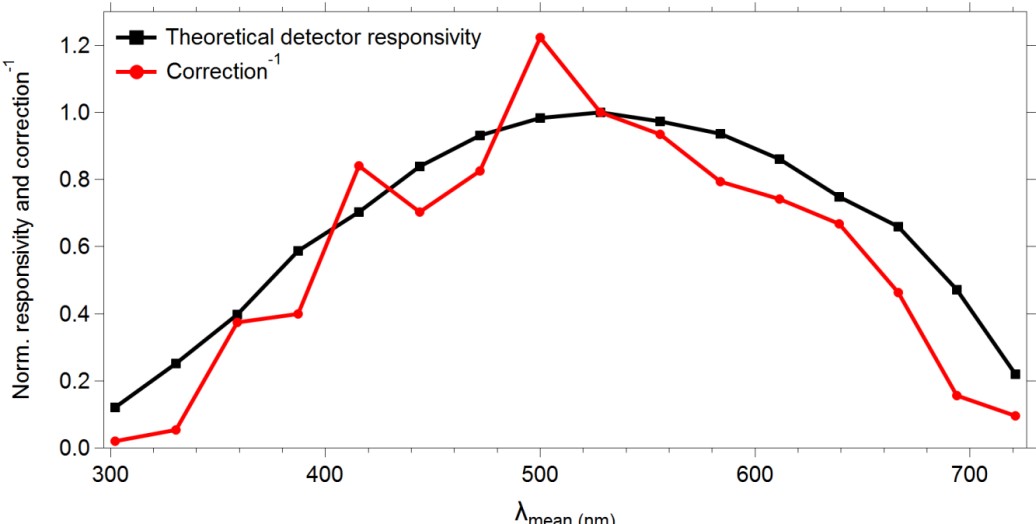

**Figure 6.** Normalized theoretical detector responsivity and spectral correction. Theoretical detector responsivity derived from measured cathode radiant sensitivity multiplied by the diffraction efficiency (as shown in Figure S8). Note that red line shows inverse of spectral correction to match detector response.

**Figure 7.** Fluorescence emission spectra of PSLs. Steady-state fluorescence signatures displayed as EEMs (left column) and spectra at Xe1 and Xe2 (middle, right columns) for: 2.07 µm purple (**a, b** and **c**, 1082 particles), 2.1 µm blue (**d, e** and **f**, 1557 particles), 2 µm green (**g, h,** and **i**, 1174 particles), and 2 µm red PSLs (**j, k,** and **l**, 1474 particles). Within EEMs: white dashed lines show SIBS excitation wavelengths ($\lambda_{ex}$ = 285 and 370 nm), grey diagonal lines indicate $1^{st}$ and $2^{nd}$ order elastic scattering bands (both bands were subtracted automatically by the Aqualog V3.6 software).

**Figure 8.** Fluorescence emission spectra of biofluorophores. EEMs (left column) and spectra at Xe1
and Xe2 wavelengths (middle and right columns) shown for: tyrosine (**a**, **b**, and **c**, 209 particles),
tryptophan (**d**, **e**, and **f**, 193 particles), NAD (**g**, **h**, and **i**, 376 particles), and riboflavin (**j**, **k**, and **l**,
205 particles). All biofluorophores were size-selected between 1 and 2 µm.



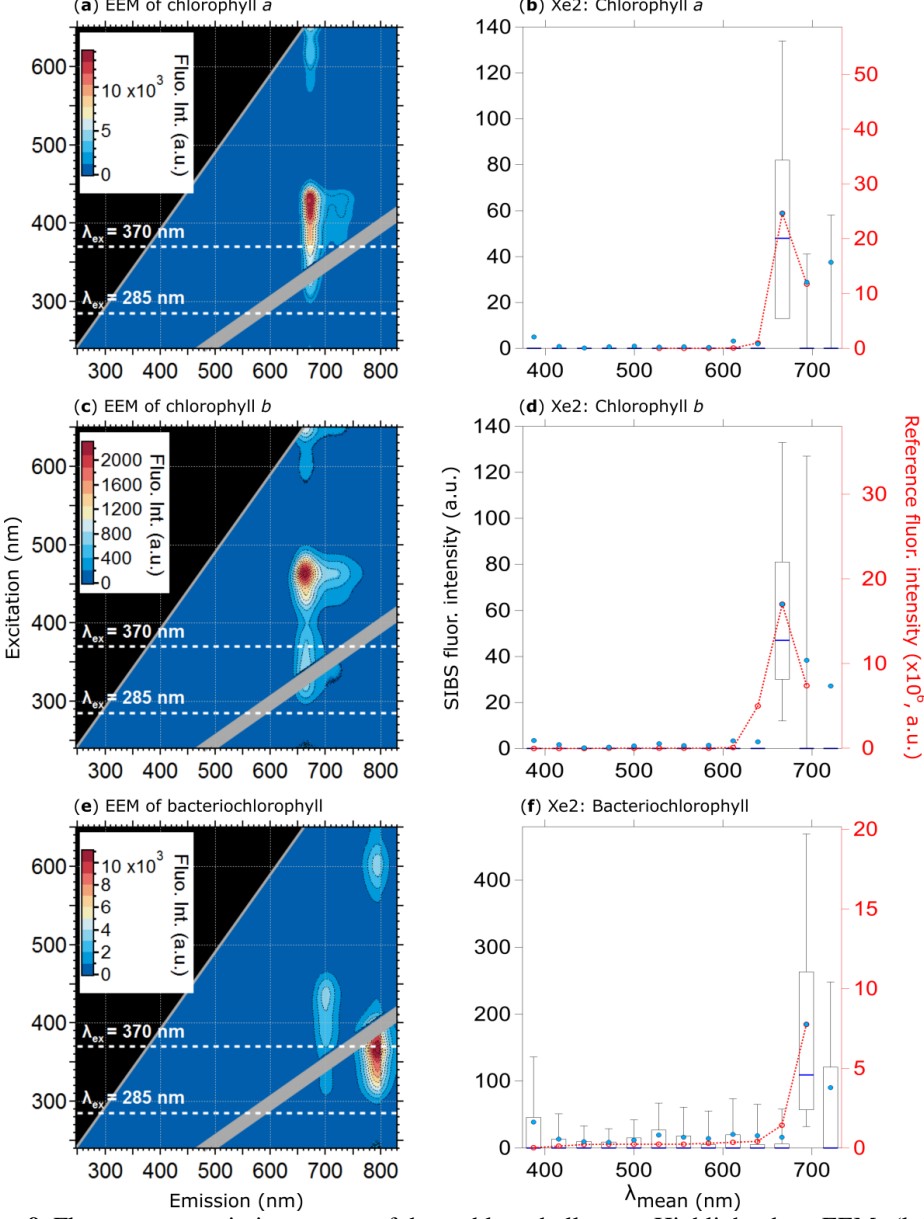

**Figure 9.** Fluorescence emission spectra of three chlorophyll types. Highlighted are EEMs (left column) and spectra at Xe2 (right columns) for: chlorophyll *a* (**a** and **b**, 370 particles), chlorophyll *b* (**c** and **d**, 585 particles), and bacteriochlorophyll (**e** and **f**, 633 particles). Size range chlorophyll *a* and *b*: 0.5 - 2 µm, size range bacteriochlorophyll: 0.5 - 1 µm. Emission spectra at Xe1 are excluded due to a fluorescence artifact caused by solved components from the polymer of the aerosolization bottles (Fig. S12).





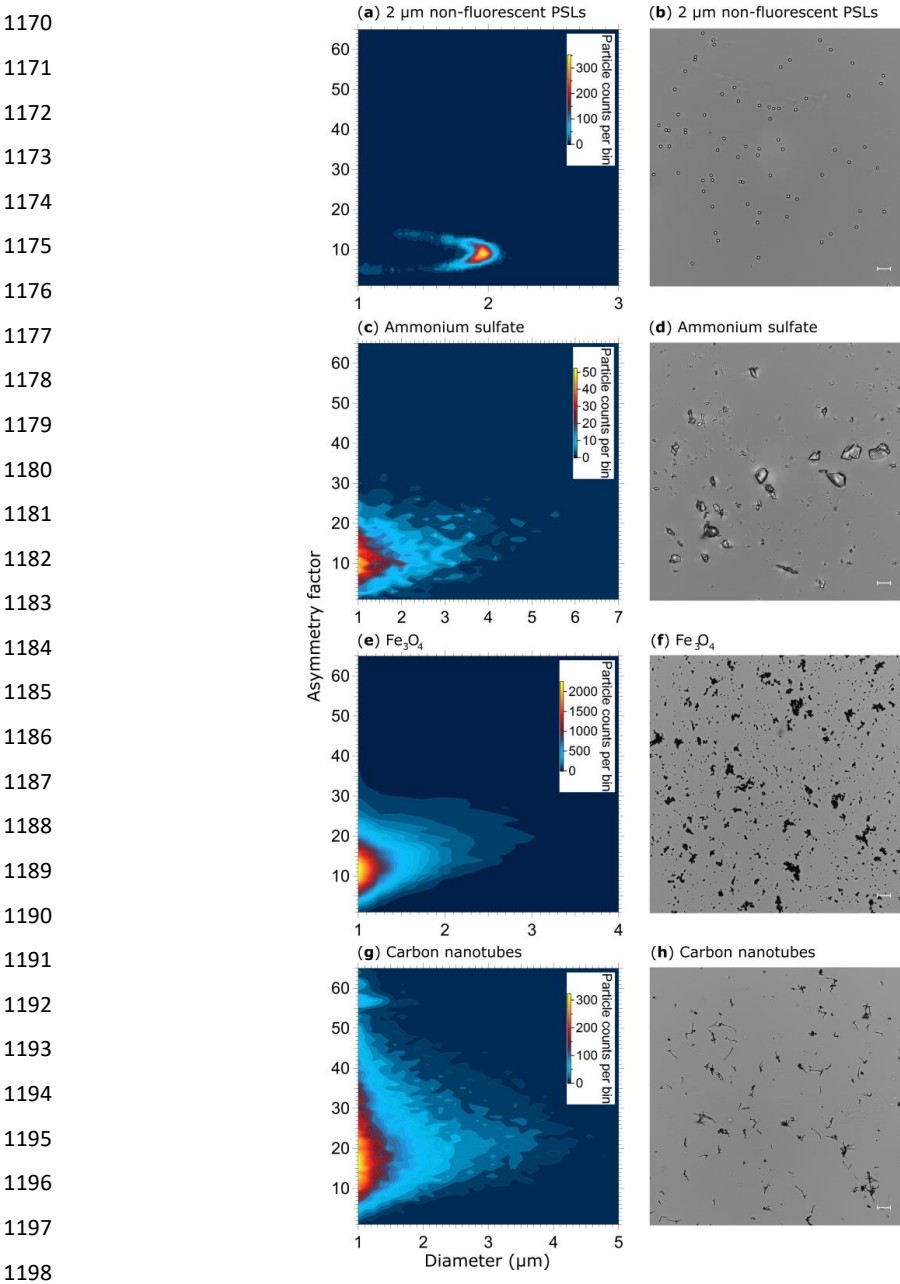

**Figure 10.** Particle asymmetry. Shown are particle density histograms (left column) and microscopy images (right column) for: 2 µm non-fluorescent PSLs (**a** and **b**, 17836 particles), ammonium sulfate (**c** and **d**, 3496 particles), Fe₃O₄ (**e** and **f**, 65097 particles), and carbon nanotubes (56949 particles, **g**). Scale bar (right column) indicates a length of 10 µm.

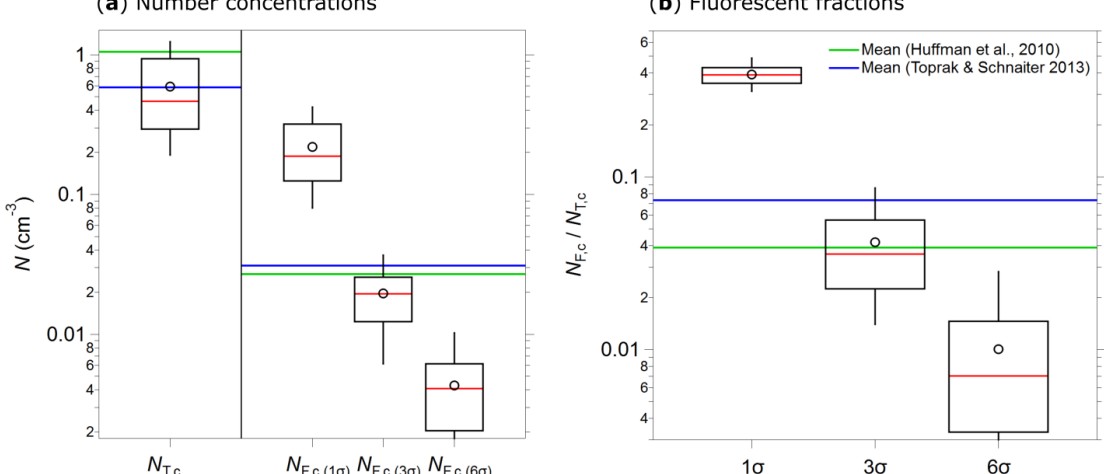

**Figure 11.** Integrated coarse particle (1-20 μm) number concentrations, measured between the 12th

and 18th of April 2018 (5 min average), for total particles ($N_{T,c}$, fluorescent and non-fluorescent) and

coarse fluorescent particles ($N_{F,c}$) after 1, 3, and 6σ SD background signal subtraction (**a**). The fluo-

rescent fractions of integrated coarse particle number concentrations ($N_{F,c} / N_{T,c}$) at 1, 3, and 6σ SD

are shown in (**b**). Median (red line), mean (black circles), boxes 75 and 25 percentile, whiskers 95

and 5 percentile (**a** and **b**). Data from Huffman et al. (2010) (green lines) and Toprak & Schnaiter,

(2013) (blue lines) were taken for comparison (**a** and **b**).