# Peer review of "Spectral Intensity Bioaerosol Sensor (SIBS): # A new Instrument for Spectrally Resolved Fluorescence Detection # of Single Particles in Real-Time"

_Atmospheric Measurement Techniques, 2018_

## Referee Comment (RC1) · Anonymous Referee #1 · 25 Nov 2018

The paper describes an instrument that should be much superior to the WIBS or UVAPS in characterizing fluorescent aerosol. The 16-channel fluorescence spectra should provide far more information for characterizing aerosol than existing WIBS instruments, while still measuring very large numbers of spectra per day. This instrument appears to be able to significantly expand our understanding of bioaerosols and other fluorescent particles in the atmosphere. The paper is clearly written. It should be published. I suggest the authors think about the following.

1. The title and abstract say, "a new instrument for . . ." Then line 144 is: "Introduced

here is a new instrument for the detection and characterization of individual particles; the Spectral Intensity Bioaerosol Sensor (SIBS, Droplet Measurement Technologies)." Then later the text says, "The SIBS was originally designed and marketed to record time-resolved fluorescence lifetime." If it has already been marketed, the use of "new" seems possibly inaccurate. How long has it been marketed? I suggest dropping "new" from the title and the text. There is no need for it. Also, a book chapter by Huffman (one of the authors of the present paper) and Santarpia, "Online Techniques for Quantification and Characterization of Biological Aerosols," in Microbiology of Aerosols eds., Anne-Marie Delort and P Amato (2017) mentions both types of SIBS (the breakdown spectroscopy SIBS and the spectral intensity SIBS). That chapter was published over a year ago and was presumably written many months before that.

2. A new and noteworthy part of this paper (maybe the most new and noteworthy part) is that the instrument is commercially available. Instruments that could do the key parts of what is done here (two fluorescence spectra each with a different excitation-wavelength is measured for each particle) have been around for some years, e.g., Huang, Pan et al., and Pan et al. But routine measurements were far from feasible by others. I suggest stating in the abstract and introduction that the instrument is built by DMT and commercially available. I suspect more people will read it if they know they could buy one. Many instruments described in papers, especially new instruments, can only be used by the researchers that built and know how to use them.

3. RE: "originally designed and marketed to record time-resolved fluorescence lifetimes". Are spectra required for measuring fluorescence lifetimes? Was the SIBS designed and marketed to measure spectra at two excitation wavelengths? I think what is meant is: It was designed and marketed to measure time- and spectrally-resolved fluorescence lifetimes.

4. Make Fig. 2 higher resolution so it can be seen in detail on a large monitor.

5. SIBS (Spark Induced Breakdown Spectroscopy) already has a meaning in the measurement of aerosol particles, either single particles or many at a time. It is confusing to see SIBS used for the name of an instrument that has nothing to do with spark induced breakdown. SIBS (original meaning) provides information similar to LIBS, i.e., elemental composition of single-particles or multiple-particles. I imagine SIBS (or LIBS) may eventually be combined with an instrument such as the SIBS described in this paper, to provide both breakdown spectra and fluorescence spectra for each particle. Since the SIBS of Konemann et al., is already marketed, and been around for a while, it is likely too late for this comment to be relevant, but I hope not. Some papers on SIBS for aerosol detection:

Spark-induced breakdown spectroscopy-based classification of bioaerosols, A J Ray Bauer and D M Sonnenfroh, 2009 IEEE International Workshop on Safety, Security & Rescue Robotics (SSRR 2009)

Preliminary correlations of feature strength in spark-induced breakdown spectroscopy of bioaerosols with concentrations measured in laboratory analyses, Shmidt, Morgan S., Bauer, Amy J. Ray, APPLIED OPTICS, 49, C101-C109 (2010) DOI: 10.1364/AO.49.00C101

Amy J.R. Hunter , Joseph R. Morency , Constance L. Senior , Steven J. Davis & Mark E. Fraser (2000) Continuous Emissions Monitoring Using Spark-Induced Breakdown Spectroscopy, Journal of the Air & Waste Management Association, 50:1, 111-117, DOI: 10.1080/10473289.2000.1046398

Continuous dust monitoring and analysis by spark induced breakdown spectroscopy, Khalaji, M., Roshanzadeh, B., Mansoori, A., Taefi, N., Tavassoli, S. H., OPTICS AND LASERS IN ENGINEERING. 50, 110-11 (2012)

---

## Referee Comment (RC2) · Crawford (Referee) · 4 Dec 2018

This paper examines the technical capabilities of the new SIBS UV-LIF bioaerosol spectrometer and describes several technical corrections and calibrations that are necessary to deliver high quality and accurate data products. As a long term WIBS user it is encouraging to see the next generation of high spectral resolution UV-LIF spectrometers that are coming to market being examined in detail early on in their lifecycle; while there is still undoubtedly still utility in broadband spectrally integrated instruments such as the WIBS for broad bioaerosol detection, it has been clear for some time now that

deeper specificity/classification requires greater spectral resolution so these technical developments are timely. The authors present a fair assessment of SIBS capability to resolve key biofluorophores and make a number of suggestions and cautions that apply to the SIBS and also UV-LIF spectrometers generally. Overall the paper is well written and the technical validation experiments are well thought out. The results and methodologies reported here will serve as a useful framework for assessing the performance of other multichannel high spectral resolution UV-LIF spectrometers which are entering circulation. I recommend publication after the following comments have been addressed.

Specific comments:

L98: Can you please check the size range reported for the WIBS-NEO. It is my understanding that the instrument sizes over the range of 0.5-30 $\mu$m.

L125: I think that a short sentence summarising some of the validation work would round this out while showing some of the limitations of the instrument/approach. A statement on how the Crawford et al. (2015) method was validated by Gosselin et al. (2016) by showing a good correlation between fungal molecular tracers and assumed fungal clusters but poor agreement between bacterial tracers and assumed bacterial clusters would contextualise this. It may also be worth commenting that the relatively high lower size limit of 0.8 $\mu$m used in this study due to instrument limitations may have impacted the latter which may potentially be alleviated by an improved lower detection limit.

L209: Can you comment further on the choice of 1$\sigma$ thresholding use here. I appreciate that the conventional wisdom used to determine the threshold for WIBS instruments may not carry over here due to the differences in the optical setup but 3$\sigma$ and 9$\sigma$ thresholds are used later in the paper when reporting ambient concentrations.

L218: This looks like it may be due to coincidence errors arising from multiple particles being present in the sample volume causing odd scattering behaviour. This is a known

problem when sampling high concentrations with forward scattering cloud probes, resulting in spectral broadening (e.g., Cooper, 1988).

L435: This is a very interesting point that is raised here about the range irradiance imbalance between xenon lamps. This confirms some of my suspicious about the utility of presenting ABC analysis in general terms without appropriate caveats or a calibration standard and I think this is worth further comment. The Hernandez et al. (2016) work showed some of the results of the issues mentioned here when they compared two WIBS-4As where there were some significantly different classifications between the two units for the same test particle. They speculated that the difference between units was due to detector gain but your results suggest that xenon intensity may significantly contribute towards the observed differences. As a follow on comment this also shows the need for a common calibration reference standard to be adopted by the UV-LIF community (e.g., Robinson et al., 2017). This potentially raises a significant challenge for UV-LIF spectrometers with increased spectral resolution as I don't know if there is likely to be a single fluorophore that will adequately cover the whole spectral range?

L517: In my experience of calibrating forward scattering cloud probes it is often common to find a dip in sizing performance in the lower region of an instruments detection range due to Mie-Lorenz resonances in the applied Mie curve exceeding the bin thresholds or the bin thresholds being relatively narrow. Mis-sizing can also be further exacerbated by the particles position in the sample area as recently demonstrated by Faber et al. (2018), however this is less likely to be an issue with SIBS/WIBS type instruments as the sample flow jet should be well constrained to the central sampling region. Given that the fit to the calibration has a slope of approximately 1 and a negligible intercept the assumed Mie curve appears to be adequate, however, should there routinely be a dip in the particle size distribution around this size this may explain why.

Technical Corrections:

Fig. 7: Can you add to the caption what the red line represents. I assume it is the

rebinned reference spectra as in Fig. 5.

Fig. S10: This would be easier to interpret if the two plots were scaled over the same x-axis range.

References:

Crawford, I., Ruske, S., Topping, D. O., and Gallagher, M. W.: Evaluation of hierarchical agglomerative cluster analysis methods for discrimination of primary biological aerosol, Atmos. Meas. Tech., 8, 4979-4991, https://doi.org/10.5194/amt-8-4979-2015, 2015.

Cooper, W.A., 1988: Effects of Coincidence on Measurements with a Forward Scattering Spectrometer Probe. J. Atmos. Oceanic Technol., 5, 823–832, https://doi.org/10.1175/1520-0426(1988)005<0823:EOCOMW>2.0.CO;2

Faber, S., French, J. R., and Jackson, R.: Laboratory and in-flight evaluation of measurement uncertainties from a commercial Cloud Droplet Probe (CDP), Atmos. Meas. Tech., 11, 3645-3659, https://doi.org/10.5194/amt-11-3645-2018, 2018.

Gosselin, M. I., Rathnayake, C. M., Crawford, I., Pöhlker, C., Fröhlich-Nowoisky, J., Schmer, B., Després, V. R., Engling, G., Gallagher, M., Stone, E., Pöschl, U., and Huffman, J. A.: Fluorescent bioaerosol particle, molecular tracer, and fungal spore concentrations during dry and rainy periods in a semi-arid forest, Atmos. Chem. Phys., 16, 15165-15184, https://doi.org/10.5194/acp-16-15165-2016, 2016.

Hernandez, M., Perring, A. E., McCabe, K., Kok, G., Granger, G., and Baumgardner, D.: Chamber catalogues of optical and fluorescent signatures distinguish bioaerosol classes, Atmos. Meas. Tech., 9, 3283-3292, https://doi.org/10.5194/amt-9-3283-2016, 2016.

Robinson, E. S., Gao, R.-S., Schwarz, J. P., Fahey, D. W., and Perring, A. E.: Fluorescence calibration method for single-particle aerosol fluorescence instruments, Atmos. Meas. Tech., 10, 1755-1768, https://doi.org/10.5194/amt-10-1755-2017, 2017.

---

## Author Comment (AC2) · 1 Feb 2019

**Response to referee comments and suggestions on amt-2018-390 by Könemann et al.**

**Manuscript format description:**

Black text shows the original referee comment, blue text shows the authors response, and red text shows quoted manuscript text. Changes to the manuscript text are shown as *italicized and underlined*. We used bracketed comment numbers for referee comments (e.g., [R2.1]) and author's responses (e.g., [A2.1]).

Line numbers refer to the discussion/review manuscript.

**Referee #2 Dr. Ian Crawford**

Received: 4 December 2018

General comment:

This paper examines the technical capabilities of the new SIBS UV-LIF bioaerosol spectrometer and describes several technical corrections and calibrations that are necessary to deliver high quality and ac- curate data products. As a long term WIBS user it is encouraging to see the next generation of high spectral resolution UV-LIF spectrometers that are coming to market being examined in detail early on in their lifecycle; while there is still undoubtedly still utility in broadband spectrally integrated instruments such as the WIBS for broad bioaerosol detection, it has been clear for some time now that deeper speci- ficity/classification requires greater spectral resolution so these technical developments are timely. The authors present a fair assessment of SIBS capability to resolve key biofluorophores and make a number of suggestions and cautions that apply to the SIBS and also UV-LIF spectrometers generally. Overall the paper is well written and the technical validation experiments are well thought out. The results and meth- odologies reported here will serve as a useful framework for assessing the performance of other multi- channel high spectral resolution UV-LIF spectrometers which are entering circulation. I recommend pub- lication after the following comments have been addressed.

Author response: We want to thank Dr. Crawford (Referee #2) for his positive assessment and construc- tive suggestions.

Specific/technical comment:

[R2.1] L98: Can you please check the size range reported for the WIBS-NEO. It is my understanding that the instrument sizes over the range of 0.5-30 μm.

[A2.1] Thanks a lot for pointing that out. The size range we stated for the WIBS-NEO originated from information we had in the beginning of 2017. Since then, DMT seem to have updated related information. The size range, within the manuscript, is now changed from ~0.3 – 100 µm to ~0.5 – 30 µm for the WIBS-NEO, according to: http://www.dropletmeasurement.com/wideband-integrated-bioaerosol-sensor-wibs-neo

[R2.2] L125: I think that a short sentence summarising some of the validation work would round this out while showing some of the limitations of the instrument/approach. A statement on how the Crawford et al. (2015) method was validated by Gosselin et al. (2016) by showing a good correlation between fungal molecular tracers and assumed fungal clusters but poor agreement between bacterial tracers and assumed bacterial clusters would contextualise this. It may also be worth commenting that the relatively high lower size limit of 0.8 μm used in this study due to instrument limitations may have impacted the latter which may potentially be alleviated by an improved lower detection limit.

[A2.2] As suggested, the following sentences have been added to round out the topic of currently used clustering approaches regarding online LIF:

(P4-5, L130-139*): "For example, it was shown for a rural forest study in Colorado that a cluster derived using WIBS-3 data, assigned to fungal spores (Crawford et al., 2015), correlated well with the mass concentration of molecular fungal tracers (e.g., arabitol and mannitol) measured with offline chemical techniques (Gosselin et al., 2016). In contrast, the clusters in the same study that were assigned to bacteria correlated only poorly with endotoxins, used as bacterial molecular tracers (Gosselin et al., 2016). This provides evidence of a limitation to using LIF instrumentation with low spectral resolution to separate or identify some PBAP types. Additionally, the bacterial cluster*

*allocation might have also been hampered in that case by the minimum detectable particle size of*

*the WIBS (~0.8 μm), resulting in a lower detection efficiency for bacteria."*

[R2.3] L209: Can you comment further on the choice of 1σ thresholding use here. I appreciate that the conventional wisdom used to determine the threshold for WIBS instruments may not carry over here due to the differences in the optical setup but 3σ and 9σ thresholds are used later in the paper when reporting ambient concentrations.

[A2.3] As pointed out by Dr. Crawford, it is currently unknown if thresholding strategies conventionally used for several WIBS models perform similar when applied to the optical setup of the SIBS. For the current manuscript, we decided to use a rather simple 1σ approach, because for the assessment of the spectral accuracy, measuring sets of homogenous particle types (PSLs, biofluorophores), the thresholding plays only a minor role. In contrast, conventional thresholding strategies were applied to a set of ambient data as a first attempt to qualitatively match SIBS results with data derived from established online LIF instruments like the WIBS and UV-APS. In this context, we added the following sentence:

(P7, L223-227): *"Optimization of the thresholding strategy is still an on-going work, for example to investigate whether the often applied 3σ threshold used for the WIBS (e.g., Gabey et al., 2010) also works well with respect to the optical setup of the SIBS. For the assessment of the accuracy of measured fluorescence emissions from reference compounds, a threshold of 1σ was used here."*

[R2.4] L218: This looks like it may be due to coincidence errors arising from multiple particles being present in the sample volume causing odd scattering behaviour. This is a known problem when sampling high concentrations with forward scattering cloud probes, resulting in spectral broadening (e.g., Cooper, 1988).

[A2.4] Thanks a lot for this hint. The stated reference might indeed be an explanation for the effects we have observed for asymmetry factor measurements with the SIBS. The following sentence was added:

(P8, L235-237): "*However, one explanation could be optical coincidences caused by high particle concentrations, resulting in multiple particles being simultaneously present within the scattering volume, as reported by Cooper (1988) using forward-scattering signatures of cloud probes.*"

[R2.5] L435: This is a very interesting point that is raised here about the range irradiance imbalance between xenon lamps. This confirms some of my suspicious about the utility of presenting ABC analysis in general terms without appropriate caveats or a calibration standard and I think this is worth further comment. The Hernandez et al. (2016) work showed some of the results of the issues mentioned here when they compared two WIBS-4As where there were some significantly different classifications between the two units for the same test particle. They speculated that the difference between units was due to detector gain but your results suggest that xenon intensity may significantly contribute towards the observed differences. As a follow on comment this also shows the need for a common calibration reference standard to be adopted by the UV-LIF community (e.g., Robinson et al., 2017). This potentially raises a significant challenge for UV-LIF spectrometers with increased spectral resolution as I don't know if there is likely to be a single fluorophore that will adequately cover the whole spectral range?

[A2.5] This observation is indeed a critical point when it comes to the interpretation of fluorescence data derived from online LIF instruments using similar optical setups. Observed differences, between similar instruments as stated in, e.g., Hernandez et al. (2016), are most likely based on the complex interaction of multiple technical components, batch-to-batch variability etc. However, if prospective experiments verify a general imbalance between xenon sources / optical filtering for the WIBS and SIBS, this issue might turn out to be a major contributor to this topic. We agree with Dr. Crawford that it is absolutely necessary to adopt a calibration standard within the online bioaerosol community. However, to the best of our knowledge, there is currently no compound available that fulfills the requirements (e.g., stability, repeatability, broad spectral range etc.) for being a standard calibrant for multi-channel, multi-excitation LIF-instruments.

Within "5. Summary and conclusions", this existing text passage briefly discuss the data interpretation issue:

(P24, L799-805): "These observations are valid not only for the SIBS, but also for the WIBS-4A and WIBS-NEO and lead to important implications for interpretation of particle data. In particular, a particle that exhibits measurable fluorescence in WIBS channel FL1, but only weak fluorescence in channel FL3 could be assigned as an "A-type" particle in one instrument but an "AC-type" particle in an instrument with slightly stronger xenon 2 irradiance. These differences in classification can be extremely important to interpretation of ambient data (Perring et al., 2015; Savage et al., 2017)."

Additionally, we added the following sentence regarding instrument intercomparisons / calibrant standards:

(P24, L794-799)*: "Additionally, alternating irradiance properties might significantly contribute to observed differences in performance of similar instrument types (e.g., Hernandez et al., 2016), expressly underlining the need for a fluorescence calibrant applicable across LIF-instruments (e.g., Robinson et al., 2017). Nevertheless, to the best of our knowledge, there is currently no standard reference available that fulfills the requirements to serve as a calibrant for multi-channel, multi-excitation LIF-instruments."*

[R2.6] L517: In my experience of calibrating forward scattering cloud probes it is often common to find a dip in sizing performance in the lower region of an instruments detection range due to Mie-Lorenz resonances in the applied Mie curve exceeding the bin thresholds or the bin thresholds being relatively narrow. Mis-sizing can also be further exacerbated by the particles position in the sample area as recently demonstrated by Faber et al. (2018), however this is less likely to be an issue with SIBS/WIBS type instruments as the sample flow jet should be well constrained to the central sampling region. Given that the fit to the calibration has a slope of approximately 1 and a negligible intercept the assumed Mie curve appears to be adequate, however, should there routinely be a dip in the particle size distribution around this size this may explain why.

[A2.6] We considered this possibility, and almost added a comment to the discussion manuscript to this effect. Looking into the Mie curves in more detail, however, we did not find a solid evidence that may serve as an explanation for the effect observed in a size range between 0.6 – 0.8 µm. Because the idea was not strongly supported and to avoid inadvertently leading readers astray, we decided to leave the issue with unknown cause.

[R2.7] Fig. 7: Can you add to the caption what the red line represents. I assume it is the rebinned reference spectra as in Fig. 5.

[A2.7] True, red dashed lines show re-binned reference spectra as stated in Fig. 5 for **c** and **d**. The caption was modified for all corresponding figures (manuscript and supplement) as requested.

[R2.8] Fig. S10: This would be easier to interpret if the two plots were scaled over the same x-axis range.

[A2.8] Within the supplement manuscript, Fig. S10 was modified as requested.

**References**

Cooper, W. A.: Effects of coincidence on measurements with a forward scattering spectrometer probe, J. Atmos. Ocean. Technol., 5(6), 823–832, 1988.

Crawford, I., Ruske, S., Topping, D. O. and Gallagher, M. W.: Evaluation of hierarchical agglomerative cluster analysis methods for discrimination of primary biological aerosol, Atmos. Meas. Tech., 8(11), 4979–4991, 2015.

Gabey, a. M., Gallagher, M. W., Whitehead, J., Dorsey, J. R., Kaye, P. H. and Stanley, W. R.: Measurements and comparison of primary biological aerosol above and below a tropical forest canopy using a dual channel fluorescence spectrometer, Atmos. Chem. Phys., 10(10), 4453–4466, doi:10.5194/acp-10-4453-2010, 2010.

Gosselin, M. I., Rathnayake, C. M., Crawford, I., Pöhlker, C., Fröhlich-Nowoisky, J., Schmer, B., Després, V. R., Engling, G., Gallagher, M., Stone, E., Pöschl, U., and Huffman, J. A.: Fluorescent bioaerosol particle, molecular tracer, and fungal spore concentrations during dry and rainy periods in a semi-arid forest, Atmos. Chem. Phys., 16(23), 15165–15184, 2016.

Hernandez, M., Perring, A. E., McCabe, K., Kok, G., Granger, G. and Baumgardner, D.: Chamber catalogues of optical and fluorescent signatures distinguish bioaerosol classes, Atmos. Meas. Tech., 9(7), 3283–3292, 2016.

Perring, A. E., Schwarz, J. P., Baumgardner, D., Hernandez, M. T., Spracklen, D. V., Heald, C. L., Gao, R. S., Kok, G., McMeeking, G. R., McQuaid, J. B. and Fahey, D. W.: Airborne observations of regional variation in fluorescent aerosol across the United States, J. Geophys. Res. Atmos., 120(3), 1153–1170, doi:10.1002/2014JD022495, 2015.

Savage, N. J., Krentz, C. E., Könemann, T., Han, T. T., Mainelis, G., Pöhlker, C. and Huffman, J. A.: Systematic characterization and fluorescence threshold strategies for the wideband integrated bioaerosol sensor (WIBS) using size-resolved biological and interfering particles, Atmos. Meas. Tech., 10(11), 4279–4302, doi:10.5194/amt-10-4279-2017, 2017.

---

## Author Response (AR1)

This document includes:

   i.    Point-by-point responses to Referee #1 and Referee #2.

  ii.    The revised manuscript (including changes, as requested by Referee #1 and #2, which are highlighted in ==yellow==. Grey highlights are for general edits.

 iii.   Manuscript supplement.

**Manuscript format description:**

Black text shows the original referee comment, blue text shows the authors response, and red text shows quoted manuscript text. Changes to the manuscript text are shown as *italicized and underlined*. We used bracketed comment numbers for referee comments (e.g., [R1.1]) and author's responses (e.g., [A1.1]). Line numbers refer to the discussion/review manuscript.

**Response to referee comments and suggestions on amt-2018-390 by Könemann et al.**

**Anonymous Referee #1**

Received: 25 November 2018

General comment:

The paper describes an instrument that should be much superior to the WIBS or UVAPS in characterizing fluorescent aerosol. The 16-channel fluorescence spectra should provide far more information for characterizing aerosol than existing WIBS instruments, while still measuring very large numbers of spectra per day. This instrument appears to be able to significantly expand our understanding of bioaerosols and other fluorescent particles in the atmosphere. The paper is clearly written. It should be published. I suggest the authors think about the following.

Author response: We want to thank Referee #1 for his/her positive and constructive assessment.

Specific/technical comment:

[R1.1] The title and abstract say, "a new instrument for . . ." Then line 144 is: "Introduced here is a new instrument for the detection and characterization of individual particles; the Spectral Intensity

Bioaerosol Sensor (SIBS, Droplet Measurement Technologies)." Then later the text says, "The SIBS was originally designed and marketed to record time-resolved fluorescence lifetime." If it has already been marketed, the use of "new" seems possibly inaccurate. How long has it been marketed? I suggest dropping "new" from the title and the text. There is no need for it. Also, a book chapter by Huffman (one of the authors of the present paper) and Santarpia, "Online Techniques for Quantification and Characterization of Biological Aerosols," in Microbiology of Aerosols eds., Anne-Marie Delort and P Amato (2017) mentions both types of SIBS (the breakdown spectroscopy SIBS and the spectral intensity SIBS). That chapter was published over a year ago and was presumably written many months before that.

[A1.1] We agree with Referee#1 and took out the word "new" from the title and abstract. It is true that the SIBS was briefly introduced within the book chapter "Online Techniques for Quantification and Characterization of Biological Aerosols" (Huffman and Santarpia, 2017). This reference is based on the same unit as discussed in amt-2018-390 and referenced by a conference poster, because no other citation was available at that time. Information stated in this book chapter was based on unpublished and non-peer-reviewed data, available because we had already been working together with Alex Huffman in 2015 with respect to the earliest version of the SIBS. Since then, the instrument underwent many modifications (hardware and software) and revisions for which the SIBS unit from 2015 and the unit in its current state are not comparable anymore.

[R1.2] A new and noteworthy part of this paper (maybe the most new and noteworthy part) is that the instrument is commercially available. Instruments that could do the key parts of what is done here (two fluorescence spectra each with a different excitation wavelength is measured for each particle) have been around for some years, e.g., Huang, Pan et al., and Pan et al. But routine measurements were far from feasible by others. I suggest stating in the abstract and introduction that the instrument is built by DMT and commercially available. I suspect more people will read it if they know they could buy one. Many instruments described in papers, especially new instruments, can only be used by the researchers that built and know how to use them.

[A1.2] As suggested by Referee#1, we added a reference to DMT within the abstract and con- clusions for clarification. The linkage between the SIBS and DMT is already given, within the introduction, in:

(P5, L155-156): "Introduced here is a new instrument for the detection and characterization of individual particles; the Spectral Intensity Bioaerosol Sensor (SIBS, Droplet Measurement Technologies)."

[R1.3] RE: "originally designed and marketed to record time-resolved fluorescence lifetimes". Are spectra required for measuring fluorescence lifetimes? Was the SIBS designed and marketed to measure spectra at two excitation wavelengths? I think what is meant is: It was designed and marketed to measure time- and spectrally-resolved fluorescence lifetimes.

[A1.3] Correct. The SIBS was originally designed to measure time- and spectrally-resolved fluorescence lifetimes at two excitation wavelengths. As suggested by Referee #1, the following sentence was changed from:

(P15, L495-496): "The SIBS was originally designed and marketed to record time-resolved fluorescence lifetime."

To (P15, L495-496): "*The SIBS was originally designed and marketed to record time- and spectrally-resolved fluorescence lifetimes at two excitation wavelengths.*"

[R1.4] Make Fig. 2 higher resolution so it can be seen in detail on a large monitor.

[A1.4] Within the current manuscript version, figures were used in lower resolution to keep file sizes as low as possible. The final version will include high resolution images and figures.

[R1.5] RE: SIBS (Spark Induced Breakdown Spectroscopy) already has a meaning in the measurement of aerosol particles, either single particles or many at a time. It is confusing to see SIBS used for the name of an instrument that has nothing to do with spark induced breakdown. SIBS (original meaning) provides information similar to LIBS, i.e., elemental composition of single-particles or multiple-particles. I imagine SIBS (or LIBS) may eventually be combined with an instrument such as the SIBS described in this paper, to provide both breakdown spectra and fluorescence spectra for each particle. Since the SIBS of Konemann et al., is already marketed, and been around for a while, it is likely too late for this comment to be relevant, but I hope not.

[A1.5] It is indeed unfortunate that two similar acronyms exist for two different instruments. We added the following sentence to hopefully avoid potential misconceptions, including references as suggested by Referee#1:

(P12, L370-373): "*To avoid potential misunderstanding, it is important to note that the SIBS described in this study is not related to spark-induced breakdown spectroscopy instrumentation, which uses the same acronym (e.g., Bauer & Sonnenfroh, 2009; Hunter et al., 2000; Khalaji et al., 2012; Schmidt & Bauer, 2010).*"

It is true that the combination of both breakdown- und fluorescence spectra on single particle scale would provide a completely new level for particle characterization. However, this topic is beyond the scope of this manuscript.

**References**

Bauer, A. J. R. and Sonnenfroh, D. M.: Spark-induced breakdown spectroscopy-based classification of bioaerosols, in Safety, Security & Rescue Robotics (SSRR), 2009 IEEE International Workshop on, pp. 1–4, IEEE., 2009.

Huffman, J. A. and Santarpia, J.: Online Techniques for Quantification and Characterization of Biological Aerosols, Microbiol. Aerosols, 83–114, 2017.

Hunter, A. J. R., Morency, J. R., Senior, C. L., Davis, S. J. and Fraser, M. E.: Continuous emissions monitoring using spark-induced breakdown spectroscopy, J. Air Waste Manage. Assoc., 50(1), 111–117, 2000.

Khalaji, M., Roshanzadeh, B., Mansoori, A., Taefi, N. and Tavassoli, S. H.: Continuous dust monitoring and analysis by spark induced breakdown spectroscopy, Opt. Lasers Eng., 50(2), 110–113, 2012.

Schmidt, M. S. and Bauer, A. J. R.: Preliminary correlations of feature strength in spark-induced breakdown spectroscopy of bioaerosols with concentrations measured in laboratory analyses, Appl. Opt., 49(13), C101–C109, 2010.

 **Response to referee comments and suggestions on amt-2018-390 by Könemann et al.**

**Referee #2 Dr. Ian Crawford**

Received: 4 December 2018

General comment:

This paper examines the technical capabilities of the new SIBS UV-LIF bioaerosol spectrometer and describes several technical corrections and calibrations that are necessary to deliver high quality and accurate data products. As a long term WIBS user it is encouraging to see the next generation of high spectral resolution UV-LIF spectrometers that are coming to market being examined in detail early on in their lifecycle; while there is still undoubtedly still utility in broadband spectrally integrated instruments such as the WIBS for broad bioaerosol detection, it has been clear for some time now that deeper specificity/classification requires greater spectral resolution so these technical develop- ments are timely. The authors present a fair assessment of SIBS capability to resolve key biofluoro- phores and make a number of suggestions and cautions that apply to the SIBS and also UV-LIF

spectrometers generally. Overall the paper is well written and the technical validation experiments are well thought out. The results and methodologies reported here will serve as a useful framework for assessing the performance of other multichannel high spectral resolution UV-LIF spectrometers which are entering circulation. I recommend publication after the following comments have been addressed.

Author response: We want to thank Dr. Crawford (Referee #2) for his positive assessment and con- structive suggestions.

Specific/technical comment:

[R2.1] L98: Can you please check the size range reported for the WIBS-NEO. It is my understanding that the instrument sizes over the range of 0.5-30 μm.

[A2.1] Thanks a lot for pointing that out. The size range we stated for the WIBS-NEO origi- nated from information we had in the beginning of 2017. Since then, DMT seem to have up- dated related information. The size range, within the manuscript, is now changed from ~0.3 –

100 μm to ~0.5 – 30 μm for the WIBS-NEO, according to: http://www.dropletmeasure- ment.com/wideband-integrated-bioaerosol-sensor-wibs-neo

[R2.2] L125: I think that a short sentence summarising some of the validation work would round this
out while showing some of the limitations of the instrument/approach. A statement on how the Craw-
ford et al. (2015) method was validated by Gosselin et al. (2016) by showing a good correlation
between fungal molecular tracers and assumed fungal clusters but poor agreement between bacterial
tracers and assumed bacterial clusters would contextualise this. It may also be worth commenting
that the relatively high lower size limit of 0.8 μm used in this study due to instrument limitations may
have impacted the latter which may potentially be alleviated by an improved lower detection limit.

[A2.2] As suggested, the following sentences have been added to round out the topic of cur-
rently used clustering approaches regarding online LIF:

(P4-5, L130-139): "For example, it was shown for a rural forest study in Colorado that a
cluster derived using WIBS-3 data, assigned to fungal spores (Crawford et al., 2015), corre-
lated well with the mass concentration of molecular fungal tracers (e.g., arabitol and mannitol)
measured with offline chemical techniques (Gosselin et al., 2016). In contrast, the clusters in
the same study that were assigned to bacteria correlated only poorly with endotoxins, used as
bacterial molecular tracers (Gosselin et al., 2016). This provides evidence of a limitation to
using LIF instrumentation with low spectral resolution to separate or identify some PBAP
types. Additionally, the bacterial cluster allocation might have also been hampered in that case
by the minimum detectable particle size of the WIBS (~0.8 μm), resulting in a lower detection
efficiency for bacteria."

[R2.3] L209: Can you comment further on the choice of 1σ thresholding use here. I appreciate that
the conventional wisdom used to determine the threshold for WIBS instruments may not carry over
here due to the differences in the optical setup but 3σ and 9σ thresholds are used later in the paper
when reporting ambient concentrations.

[A2.3] As pointed out by Dr. Crawford, it is currently unknown if thresholding strategies con-
ventionally used for several WIBS models perform similar when applied to the optical setup of
the SIBS. For the current manuscript, we decided to use a rather simple 1σ approach, because
for the assessment of the spectral accuracy, measuring sets of homogenous particle types
(PSLs, biofluorophores), the thresholding plays only a minor role. In contrast, conventional
thresholding strategies were applied to a set of ambient data as a first attempt to qualitatively
match SIBS results with data derived from established online LIF instruments like the WIBS
and UV-APS. In this context, we added the following sentence:

(P7, L223-227): *"Optimization of the thresholding strategy is still an on-going work, for example to investigate whether the often applied 3σ threshold used for the WIBS (e.g., Gabey et al., 2010) also works well with respect to the optical setup of the SIBS. For the assessment of the accuracy of measured fluorescence emissions from reference compounds, a threshold of 1σ was used here."*

[R2.4] L218: This looks like it may be due to coincidence errors arising from multiple particles being present in the sample volume causing odd scattering behaviour. This is a known problem when sampling high concentrations with forward scattering cloud probes, resulting in spectral broadening (e.g., Cooper, 1988).

[A2.4] Thanks a lot for this hint. The stated reference might indeed be an explanation for the effects we have observed for asymmetry factor measurements with the SIBS. The following sentence was added:

(P8, L235-237): "*However, one explanation could be optical coincidences caused by high particle concentrations, resulting in multiple particles being simultaneously present within the scattering volume, as reported by Cooper (1988) using forward-scattering signatures of cloud probes.*"

[R2.5] L435: This is a very interesting point that is raised here about the range irradiance imbalance between xenon lamps. This confirms some of my suspicious about the utility of presenting ABC analysis in general terms without appropriate caveats or a calibration standard and I think this is worth further comment. The Hernandez et al. (2016) work showed some of the results of the issues mentioned here when they compared two WIBS-4As where there were some significantly different classifications between the two units for the same test particle. They speculated that the difference between units was due to detector gain but your results suggest that xenon intensity may significantly contribute towards the observed differences. As a follow on comment this also shows the need for a common calibration reference standard to be adopted by the UV-LIF community (e.g., Robinson et al., 2017). This potentially raises a significant challenge for UV-LIF spectrometers with increased spectral resolution as I don't know if there is likely to be a single fluorophore that will adequately cover the whole spectral range?

[A2.5] This observation is indeed a critical point when it comes to the interpretation of fluorescence data derived from online LIF instruments using similar optical setups. Observed differences, between similar instruments as stated in, e.g., Hernandez et al. (2016), are most likely based on the complex interaction of multiple technical components, batch-to-batch variability etc. However, if prospective experiments verify a general imbalance between xenon sources / optical filtering for the WIBS and SIBS, this issue might turn out to be a major contributor to this topic. We agree with Dr. Crawford that it is absolutely necessary to adopt a calibration standard within the online bioaerosol community. However, to the best of our knowledge, there is currently no compound available that fulfills the requirements (e.g., stability, repeatability, broad spectral range etc.) for being a standard calibrant for multi-channel, multi-excitation LIF-instruments.

Within "5. Summary and conclusions", this existing text passage briefly discuss the data interpretation issue:

(P24, L799-805): "These observations are valid not only for the SIBS, but also for the WIBS-4A and WIBS-NEO and lead to important implications for interpretation of particle data. In particular, a particle that exhibits measurable fluorescence in WIBS channel FL1, but only weak fluorescence in channel FL3 could be assigned as an "A-type" particle in one instrument but an "AC-type" particle in an instrument with slightly stronger xenon 2 irradiance. These differences in classification can be extremely important to interpretation of ambient data (Perring et al., 2015; Savage et al., 2017)."

Additionally, we added the following sentence regarding instrument intercomparisons / calibrant standards:

(P24, L794-799): "Additionally, alternating irradiance properties might significantly contribute to observed differences in performance of similar instrument types (e.g., Hernandez et al., 2016), expressly underlining the need for a fluorescence calibrant applicable across LIF-instruments (e.g., Robinson et al., 2017). Nevertheless, to the best of our knowledge, there is currently no standard reference available that fulfills the requirements to serve as a calibrant for multi-channel, multi-excitation LIF-instruments."

[R2.6] L517: In my experience of calibrating forward scattering cloud probes it is often common to find a dip in sizing performance in the lower region of an instruments detection range due to Mie-

Lorenz resonances in the applied Mie curve exceeding the bin thresholds or the bin thresholds being relatively narrow. Mis-sizing can also be further exacerbated by the particles position in the sample area as recently demonstrated by Faber et al. (2018), however this is less likely to be an issue with SIBS/WIBS type instruments as the sample flow jet should be well constrained to the central sampling region. Given that the fit to the calibration has a slope of approximately 1 and a negligible intercept the assumed Mie curve appears to be adequate, however, should there routinely be a dip in the particle size distribution around this size this may explain why.

[A2.6] We considered this possibility, and almost added a comment to the discussion manuscript to this effect. Looking into the Mie curves in more detail, however, we did not find a solid evidence  that may serve as an explanation for the effect observed in a size range between 0.6 – 0.8 µm. Because the idea was not strongly supported and to avoid inadvertently leading readers astray, we decided to leave the issue with unknown cause.

[R2.7] Fig. 7: Can you add to the caption what the red line represents. I assume it is the rebinned reference spectra as in Fig. 5.

[A2.7] True, red dashed lines show re-binned reference spectra as stated in Fig. 5 for **c** and **d**. The caption was modified for all corresponding figures (manuscript and supplement) as requested.

[R2.8] Fig. S10: This would be easier to interpret if the two plots were scaled over the same x-axis range.

[A2.8] Within the supplement manuscript, Fig. S10 was modified as requested.

**References**

[revised manuscript text omitted]

 Asymmetry Factor
 Fluorescence spectra | Particle size
 Asymmetry Factor
 Integrated fluorescence in 3 channels | Particle size
 Asymmetry Factor
 Integrated fluorescence in 3 channels |
| **Particle size range** | ~0.3 – 100 µm |  *0.5* –  *30* µm | ~0.5 – 20 µm |
| **Maximum concentration** | ~2 x $10^4$ particles/L | ~2 x $10^4$ particles/L | ~2 x $10^4$ particles/L |
| **Fluorescence excitation** | $\lambda_{ex}$= 285 and $\lambda_{ex}$= 370 nm | $\lambda_{ex}$= 280 and $\lambda_{ex}$= 370 nm | $\lambda_{ex}$= 280 and $\lambda_{ex}$= 370 nm |
| **Fluorescence emission** | $\lambda_{mean}$= 302 – 721 nm
 (16-channel PMT) | $\lambda_{em}$= 310-400 nm and
 $\lambda_{em}$= 420-650 nm | $\lambda_{em}$= 310-400 nm and
 $\lambda_{em}$= 420-650 nm |
| **Flow rate** | Sample flow:~0.3 l/min
 Sheath flow:~2.2 l/min
 (re-circulating) | Sample flow:~0.3 l/min
 Sheath flow:~2.2 l/min
 (re-circulating) | Sample flow:~0.3 l/min
 Sheath flow:~2.2 l/min
 (re-circulating) |
| **Laser** | 785 nm diode laser, 55 mW | 635 nm diode laser, 15 mW | 635 nm diode laser, 12 mW |
| **Pump** | Diaphragm pump | Diaphragm pump | Diaphragm pump |
| **Power requirements** | 200 W, 90 - 230 VAC | 150 W, 90 - 230 VAC | 150 W, 90 - 230 VAC |
| **Weight (kg)** | 20.1 | 12.5 | 13.6 |
| **Dimension W x L x H (cm)** | 42.5 x 61.5 x 23.5 | 45.1 x 36.2 x 24.1 | 30.4 x 38.2 x 17.1 |

**Table 3.** Asymmetry factor (AF) values for reference particles. Values are based on the mean of a

Gaussian fit applied onto each particle histogram (see also Fig. 10), including $1\sigma$ SD.

[revised manuscript text omitted]

*Red dashed line and markers (right axes): averaged and re-binned reference spectra.*

[Figure]

**Figure S10.** Dry vs. solv*ateded*. Shown are reference spectra for NAD (**a**) and riboflavin (**b**) in dry and solv*ateded* state. Data coinciding with 2nd order elastic scattering were removed (*a and* **b**, red solid line). Peak maxima: NAD (dry): ~448 nm, NAD (solv*ateded*): ~463 nm, riboflavin (dry): ~572 nm, riboflavin (solv*ateded*): ~535 nm.

[Figure]

**Figure S11.** Fluorescence spectra of different chlorophyll types. Shown are reference spectra for chlorophyll *a*, *b*, and bacteriochlorophyll at $\lambda_{ex}$ = 370 nm.

[Figure]

**Figure S12.** Fluorescence spectra of ethanol artefact. Highlighted are fluorescence spectra of bacteriochlorophyll at Xe1 (**a**) and uncorrected spectra of ethanol, after being vortexed for 15 min in nebulizer plastic bottles, at Xe1 (**b**) and Xe2 (**c**). *Red dashed lines and markers (right axes;): averaged and re-binned reference spectra.* Since no distinct fluorescence signal is detectable at Xe2 (**c**), the fluorescence emission of chlorophyll *a*, *b* and bacteriochlorophyll is considered to be unaffected.

[Figure]

**Figure S13.** Particle asymmetry of ultrapure water droplets (163178 particles) displayed as particle density histogram.

[Figure]

**Figure S14.** Exemplary fluorescence spectra of single ambient particles at Xe1 (**a**) and Xe2

(**b**).